# RiboFlow: Conditional *De Novo* RNA Co-Design via Synergistic Flow Matching

**Runze Ma**[1,4,*], **Zhongyue Zhang**[1,*], **Zichen Wang**[1], **Chenqing Hua**[2],
**Jiahua Rao**[3], **Zhuomin Zhou**[1], **Shuangjia Zheng**[1,†]

[1] Shanghai Jiao Tong University [2] Yale University [3] Sun Yat-Sen University
[4] Lingang Laboratory * Equal contribution † Corresponding Author

## Abstract

Ribonucleic acid (RNA) binds to molecules to achieve specific biological functions. While generative models are advancing biomolecule design, existing methods for designing RNA that target specific ligands face limitations in capturing RNA's conformational flexibility, ensuring structural validity, and overcoming data scarcity. To address these challenges, we introduce RiboFlow, a synergistic flow matching model to co-design RNA structures and sequences based on target molecules. By integrating RNA backbone frames, torsion angles, and sequence features in an unified architecture, RiboFlow explicitly models RNA's dynamic conformations while enforcing sequence-structure consistency to improve validity. Additionally, we curate RiboBind, a large-scale dataset of RNA-molecule interactions, to resolve the scarcity of high-quality structural data. Extensive experiments reveal that RiboFlow not only outperforms state-of-the-art RNA design methods by a large margin but also showcases controllable capabilities for achieving high binding affinity to target ligands. Our work bridges critical gaps in controllable RNA design, offering a framework for structure-aware, data-efficient generation.

## 1 Introduction

Ribonucleic acid (RNA) is a programmable biomolecule that achieves precise molecular recognition through its dynamic three-dimensional structure [44], enabling applications in catalysis, biosensing, and therapeutic targeting [43, 7]. Advances in computational tools, exemplified by AlphaFold3 [1], have revolutionized biomolecular structure prediction, while generative models [27, 48, 21, 42] now pioneer the design of *de novo* biomolecules with specific binding properties. RNA, with its structural flexibility at the tertiary level, ease of chemical synthesis in laboratory settings, and low immunogenicity in biological systems, stands out as a promising candidate for therapeutic drugs and biochemical reagents [37, 12]. However, existing methods for designing RNAs that bind specific small molecules—critical for therapeutic and diagnostic applications—face unresolved challenges at the intersection of data availability, interaction modeling, and structural validity.

Recent work has laid foundations for RNA design. Tools like RNAiFold [13] and gRNAde [24] generate sequences matching predefined secondary or tertiary structures, while RNA-FrameFlow [3], MMDiff [31], and RNAFlow [32] focus on backbone generation. Yet, designing RNA for small-molecule targeting remains an open problem due to three gaps: (1) the inherent conformational flexibility of RNA requires the simultaneous and consistent consideration of both its structure and sequence [15]; (2) existing models lack explicit conditioning on ligand geometry, limiting their ability to capture RNA-ligand binding dynamics; and (3) the scarcity of RNA-ligand structural data restricts the scalability and generalizability of data-driven approaches.

To bridge these gaps, we identify three key challenges. First, ensuring generated RNAs satisfy both binding specificity and biophysical validity requires co-designing sequence and structure in a

synergistic framework. Second, modeling the structural flexibility of RNA, especially its torsion angles and backbone dynamics, while maintaining sequence-structure compatibility, remains a complex problem involving both geometric and thermodynamic considerations. Third, the absence of large-scale, standardized RNA-ligand interaction datasets hinders training robust generative models.

We try to address these challenges. For model design, we introduce **RiboFlow**, a synergistic flow matching model for *de novo* RNA discrete sequence and continuous structure co-design. By conditioning on ligand geometry and leveraging RNA backbone frames, torsion angles, and sequence features, RiboFlow models conformational flexibility while enforcing sequence-structure consistency. A novel co-design pre-training strategy is proposed to further enhance geometric awareness by distilling structural priors from RNA crystal structures. To overcome data scarcity, we introduce **RiboBind**, a comprehensive dataset of RNA-ligand complexes systematically curated from the PDB database, comprising 1,591 RNA-ligand complexes and 3,012 RNA-ligand pairs.

Our contributions are: (i) **Task Formulation**: We propose a first-of-its-kind synergistic flow-matching framework for ligand-conditioned *de novo* RNA design. The model incorporates torsion angle and backbone frame modeling, enabling sequence-structure co-design for specified ligands while offering controllable ligand-binding specificity. (ii) **Dataset**: We present RiboBind, a large standardized RNA-ligand interaction benchmark, enabling data-driven RNA discovery. (iii) **Evaluation**: We develop a multi-faceted pipeline assessing structural validity and binding affinity (via docking and scoring). Experimental results demonstrate that RiboFlow outperforms state-of-the-art RNA design methods by a large margin (e.g., achieving a 2.2-fold improvement in the AF3 binding metric and a 50% increase in validity), but also showcases controllable capabilities for achieving high binding affinity to target ligands. We anticipate this work advancing RNA design toward structure-aware, ligand-conditioned design, with promising applications in therapeutics and synthetic biology.

## 2 Related Work

### 2.1 RNA Design

Currently, RNA design can be broadly categorized into two main approaches: sequence-based and structure-based methods. *Sequence-based design* primarily aims to address the RNA inverse folding problem, which involves designing an RNA sequence that folds into a desired RNA structure. While early efforts [13, 46, 9, 34] largely focus on RNA secondary structure information, recent studies [38, 24, 22, 44] begin to explore the use of RNA 3D structural information to guide sequence design. On the other hand, *structure-based design*, still in its early stages, encompasses approaches such as RNA backbone design [3], which enables the creation of RNA with specific structures. Another approach is RNA-protein co-design [31, 32], which focuses on designing RNA and protein components by accounting for their interactions. However, these methods are not capable of designing ligand-targeting RNAs, limiting their applications in broader diagnostic and synthetic biology scenarios.

### 2.2 Flow Matching

Flow matching [28, 29], an emerging generative modeling approach, is increasingly recognized as a compelling alternative to traditional generative models [26, 18, 20]. It has extensive applications in both the computer vision and natural language processing fields [10, 16, 14]. Within the realm of biomolecular design, the focus of this paper, researchers are actively exploring the application of flow matching for diverse molecular design tasks. For example, several studies [50, 47, 5] utilize SE(3)-equivariant flow models to design small molecules and proteins, while others [23, 6] integrate protein sequence for protein design, or leverage multiple sequence alignment (MSA) coevolutionary information [21, 42] to design enzymes and antibodies. In contrast to these methods, RiboFlow integrates RNA-specific structural priors with sequence-structure consistency through synergistic flow matching, thereby deriving the benefits of both data-efficient representations for RNA inputs.

## 3 Preliminary

**RNA Backbone Parameterization.** As illustrated in Figure 1, an RNA has a backbone made of alternating phosphate groups ($P$, $OP1$, $OP2$, $O5'$) and the sugar ribose ($C1'$ - $C5'$, $O2'$, $O3'$, $O4'$). A nitrogen atom, located at the base attachment site ($N9$ of purines or $N1$ of pyrimidines), is often

included in modeling. Utilizing recent RNA parameterization methods [31, 3], we construct a local coordinate system for each nucleotide based on the $C4'$, $C3'$, and $O4'$ atoms of the ribose. The positions of the remaining atoms are then determined by a set of eight torsion angles, $\Phi = \{\phi_i\}_{i=1}^{8}$, where $\phi_i \in \mathbb{R}^2$, along with bond lengths and bond angles [17]. This parameterized approach can effectively captures the conformational flexibility of RNA.

**Notations and Problem Formulation.** We consider a binding system comprising an RNA-ligand pair $\mathcal{C} = \{\mathcal{T}, \mathcal{M}\}$, where the RNA, denoted by $\mathcal{T}$, consists of $N_t$ nucleotides, and the ligand, denoted by $\mathcal{M}$, consists of $N_m$ atoms. The 3D geometry of the ligand is represented as $\mathcal{M} = \{(a^{(i)}, b^{(i)})\}_{i=1}^{N_m}$, where $a^{(i)} \in \mathbb{R}^{n_m}$ indicate the atom type ($n_m$ is the total number of possible atom types), and $b^{(i)} \in \mathbb{R}^3$ denotes its 3D Cartesian coordinates. The RNA is represented by the backbone atoms of each nucleotide, and is parameterized as a set of nucleotide blocks $T^{(i)} = \{x^{(i)}, r^{(i)}, c^{(i)}\}$. Here, $x^{(i)} \in \mathbb{R}^3$ is the position of the $C4'$ atom of the $i$-th nucleotide, $r^{(i)} \in \mathrm{SO}(3)$ is a rotation matrix defining the orientation of the local frame formed by $C3' - C4' - O4'$

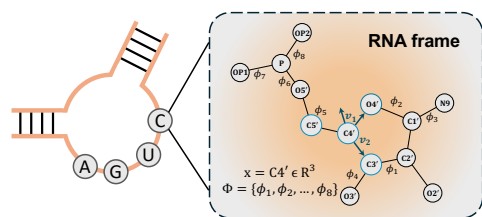

Figure 1: RNA backbone parameterization. Using the $C4'$ atom as the origin, a local coordinate system is established using $v_1$ and $v_2$ defined by vectors along the $C4'$-$O4'$ and $C4'$-$C3'$ bonds via the Gram-Schmidt process.

relative to a global reference frame, and $c^{(i)} \in \{A, C, G, U\}$ represents the nucleotide type. The complete RNA structure is represented by $\mathcal{T} = \{T^{(i)}\}_{i=1}^{N_t}$. Our goal is to develop a probabilistic model that learns the conditional distribution $p(\mathcal{T}|\mathcal{M})$, *i.e.*, generating an RNA $\mathcal{T}$ conditioned on a target ligand structure $\mathcal{M}$.

# 4 Methods

In this section, we present RiboFlow, a flow matching framework for *de novo* RNA design conditioned on target small molecules. RiboFlow operates in two key stages: *initially*, it learns a distribution of structurally plausible RNAs during a synergistic co-design pre-training stage, establishing priors for molecule-conditioned generation. *Subsequently*, it is trained to generate high-affinity RNAs given a specific target molecule. For clarity, a comprehensive overview of the flow matching is discussed in the Appendix D, with mathematical tools and proofs used in our methodology. The architecture of RiboFlow is illustrated in Figure 2.

## 4.1 Overview

As mentioned in the preliminaries, the $i$-th nucleotide in an RNA can be parameterized as $T^{(i)} = \{x^{(i)}, r^{(i)}, c^{(i)}\}$. We use time $t = 1$ to represent the target data (*i.e.*, the real RNA data $\mathcal{T}_1$), and $t = 0$ to represent noise data (*i.e.*, $\mathcal{T}_0$). Inspired by recent work of [47, 6], the conditional flow $p_t(\mathcal{T}_t|\mathcal{T}_1)$ in the RNA generation process for a time step $t \in [0, 1]$ can be expressed as:

$$p_t(\mathcal{T}_t|\mathcal{T}_1) = \prod_{i=1}^{N_t} p_t(x_t^i|x_1^i) p_t(r_t^i|r_1^i) p_t(c_t^i|c_1^i), \qquad (1)$$

where $N_t$ denotes the length of the RNA sequence. The above formula allows us to synergistically consider the probability distributions of the three parts (translation, rotation, and nucleotide type) during training and sampling, thereby modeling the RNA generation process into the SE(3) space and the nucleotide type space, respectively.

## 4.2 RiboFlow on SE(3)

We model RNA structure in SE(3) space using a set of structure frames, denoted as $\mathbf{F} = \{(x^{(i)}, r^{(i)})\}_{i=1}^{N_t}$, where $x^{(i)} \in \mathbb{R}^3$ represents the translation and $r^{(i)} \in \mathrm{SO}(3)$ represents the rotation of the $i$-th nucleotide. We define a forward process that transforms an initial noisy frame set $F_0 \sim p_0(F_0)$ to a target structure frame set $F_1 \sim p_1(F_1)$. A continuous flow $F_t$ between $F_0$ and $F_1$

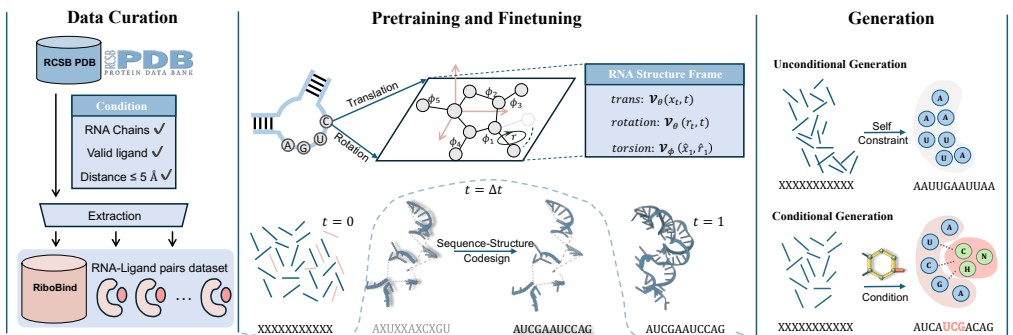

Figure 2: Framework of RiboFlow. We start by collecting and constructing RiboBind, a large-scale RNA-ligand interaction dataset, from the RCSB PDB database. RiboFlow is then pretrained on RNAsolo using a sequence-structure synergistic co-design strategy to enhance its geometric awareness. Next, the model is fine-tuned with RiboBind, enabling it to design RNAs with high predicted affinity under ligand constraints. During inference, RiboFlow generates structurally valid and high-affinity RNAs tailored to different ligands. In the figure, "X" represents uncertain nucleotides.

is constructed by interpolating on the SE(3) manifold:

$$F_t = \exp_{F_0}\left(t \cdot \log_{F_0}(F_1)\right). \tag{2}$$

Here, $\exp(\cdot)$ and $\log(\cdot)$ are the exponential and logarithmic maps respectively, enabling movement along the curved SE(3) manifold. Since the SE(3) can be decomposed into independent translations and rotations, we can also obtain the closed-form interpolations [47] for $\mathbb{R}^3$ and SO(3) separately:

$$x_t = (1-t)x_0 + tx_1; \tag{3}$$

$$r_t = \exp_{r_0}(t \cdot \log_{r_0}(r_1)), \tag{4}$$

where $x_0$ sampled from $\mathcal{N}(0, \mathbf{I})$, and $r_0$ sampled uniformly from the rotation group SO(3), denoted as $\mathcal{U}(SO(3))$. Based on these interpolations, we can derive the conditional vector fields $u_t$ for the translation and rotation components respectively [39] due to their simple nature:

$$u_t(x_t^{(i)}|x_1^{(i)}, x_0^{(i)}) = x_1^{(i)} - x_0^{(i)}; \tag{5}$$

$$u_t(r_t^{(i)}|r_1^{(i)}, r_0^{(i)}) = \log_{r_t^{(i)}}(r_1^{(i)}). \tag{6}$$

Hence, we leverage an SE(3)-equivariant neural network $\boldsymbol{v_\theta}(\cdot)$ to regress the conditional vector fields at time $t$. For $N_t$ structure frames, the loss to train the conditional flow matching for translation can be written as follows:

$$\mathcal{L}_{\text{trans}} = \mathbb{E}_{\substack{t \sim \mathcal{U}(0,1), p_1(x_1), \\ p_0(x_0), p_t(x_t|x_0, x_1)}} \sum_{i=1}^{N_t} \left\| \boldsymbol{v_\theta}^{(i)}(x_t, t) - x_1^{(i)} + x_0^{(i)} \right\|_{\mathbb{R}^3}^2, \tag{7}$$

and the loss to train rotation conditional flow matching is:

$$\mathcal{L}_{\text{rot}} = \mathbb{E}_{\substack{t \sim \mathcal{U}(0,1), p_1(r_1), \\ p_0(r_0), p_t(r_t|r_0, r_1)}} \sum_{i=1}^{N_t} \left\| \boldsymbol{v_\theta}^{(i)}(r_t, t) - \frac{\log_{r_t^{(i)}}(r_1^{(i)})}{1-t} \right\|_{\text{SO(3)}}^2. \tag{8}$$

At the same time, according to Equations 5 and 6, we can obtain the predicted structure frame $\hat{F}_1 = \{(\hat{x}_1^{(i)}, \hat{r}_1^{(i)})\}_{i=1}^{N_t}$ given the corrupted structure frame $F_t$.

### 4.3 RiboFlow on Torsion Angles

The conformational flexibility of the RNA backbone is largely determined by its 8 torsion angles. These angles, $\phi \in \Phi$, which are elements of $\mathbb{R}^2$, require appropriate constraints to generate physically plausible RNA structures. Existing studies have explored modeling torsion angles on the torus [27] and demonstrated promising experimental results. However, such an approach may increase computational

complexity when regressing conditional vector fields of torsion angles. Insipired by Anand *et al.* [3], we introduce a torsion angle prediction module, denoted as $\boldsymbol{v}_{\boldsymbol{\phi}}(\cdot)$. This module employs a shallow ResNet architecture [19], which is first implemented in the structure prediction module of AlphaFold2 [25] and has been widely adopted in recent works. By utilizing the predicted structural frames $\hat{F}_1 = \{(\hat{x}_1^{(i)}, \hat{r}_1^{(i)})\}_{i=1}^{N_t}$, the module can predict the ground-truth torsion angles $\Phi_1$ directly. The process can be formulated as follows:

$$\hat{\Phi}_1^{(i)} = \boldsymbol{v}_{\boldsymbol{\phi}}(\hat{x}_1^{(i)}, \hat{r}_1^{(i)}). \tag{9}$$

Here, $\hat{\Phi}_1^{(i)}$ represents the predicted set of 8 torsion angles for the $i$-th nucleotide. The loss between the predicted and true torsion angles can be formulated as:

$$\mathcal{L}_{\text{tors}} = \frac{1}{8N_t} \sum_{i=1}^{N_t} \sum_{\phi \in \Phi_1^{(i)}, \hat{\phi} \in \hat{\Phi}_1^{(i)}} \left\| \hat{\phi} - \phi \right\|_{\mathbb{R}^2}^2. \tag{10}$$

### 4.4 RiboFlow on Nucleotide Type

The intricate interplay between sequence and structure in RNA molecules necessitates a generation process that can effectively capture and leverage their mutual constraints. To address this, we introduce a synergistic flow matching approach that integrates the semantic information of the sequence with the structural flexibility inherent to RNA. This method not only constrains the generation process but also ensures that the generated sequences are biologically plausible and functionally relevant. For an RNA sequence $c = \{c^{(i)}\}_{i=1}^{N_t}$, we define $c_t^{(i)} \sim p(s_t^{(i)})$, where $s_t^{(i)}$ is a probability vector representing the distribution over nucleotide types, following a multinomial distribution.

Following the work of [6], we employ a conditional flow to linearly interpolate from a uniform prior to $x_1^{(i)}$. This requires that the probability vector satisfies $s_1 = \text{onehot}(c_i)$ and $s_0 = (\frac{1}{4}, \ldots, \frac{1}{4})$. The conditional flow of the probability vector is then given by:

$$s_t = ts_1 + (1 - t)s_0, \tag{11}$$

and the corresponding conditional vector field is:

$$u_t(s|s_0, s_1) = s_1 - s_0. \tag{12}$$

We use the neural network $\boldsymbol{v}_{\boldsymbol{\theta}}(\cdot)$ to predict the vector field of nucleotide types. The network's training objective is optimized by minimizing the following loss:

$$\mathcal{L}_{\text{type}} = \mathbb{E}_{\substack{t \sim \mathcal{U}(0,1), p_1(s_1), \\ p_0(s_0), p_t(s_t|s_1, s_0)}} \sum_{i=1}^{N_t} \text{CE}(s_t^{(i)} + (1 - t)\boldsymbol{v}_{\boldsymbol{\theta}}(s_t^{(i)}), s_1^{(i)}), \tag{13}$$

where $\text{CE}(\cdot)$ is the cross-entropy function. This loss directly measures the difference between the true probability distribution and the inferred distribution for the RNA sequence.

### 4.5 Distance-Aware Ligand Guided RNA Generation

While the preceding sections detailed the unconditional RNA generation through the synergistic flow matching, it is still necessary to incorporate ligand information during the training and sampling phases to guide the model toward producing RNA with specific ligand binding affinity. Therefore, we introduce a hierarchical RNA-ligand interaction module that explicitly incorporates ligand information into the RNA generation process.

Given a ligand $\mathcal{M} = \{(a^{(j)}, b^{(j)})\}_{j=1}^{N_m}$, a two-stage distance-aware structure refinement process is designed to implicitly optimize the binding free energy by modeling 3D RNA-ligand interactions. In the first stage, a multi-layer perceptron is employed to learn the embedding $h_a$ for atoms in the ligand. Subsequently, an invariant point attention (IPA) module predicts the initial RNA structural frame $\hat{F}_1 = \{(\hat{x}_1^{(i)}, \hat{r}_1^{(i)})\}_{i=1}^{N_t}$ without considering ligand interaction. Next, We compute the spatial interaction feature $h_b$ between the $i$-th RNA backbone atom ($i \in [1, N_t]$) and the $j$-th ligand atom ($j \in [1, N_m]$) as:

$$h_b = \exp\left(-\gamma \|\hat{x}_1^{(i)} - b^{(j)}\|^2\right), \tag{14}$$

where $\gamma$ is a scaling factor controlling the spatial sensitivity. In the second stage, the IPA module refines the structure using $h_a$ and $h_b$, yielding the post-interaction RNA frame:

$$\tilde{F}_1 = \text{IPA}(\hat{F}_1, h_a, h_b). \tag{15}$$

This two-stage process produces a refined RNA structure conditioned on the specific ligand, enabling conditional flow matching optimization under ligand constraints.

## 4.6 Training and Inference.

**Training.** To fully leverage ligand information as conditional inputs, we incorporate the entire RNA-ligand complex as input. Specifically, we sample $x_t$, $r_t$, and $c_t$ alongside the ligand information through the defined conditional probability paths. Consequently, the complete binding complex at time $t$ serves as input to the vector field $v_t(\cdot)$, represented as $v_t(\cdot \mid \mathcal{T} \cup \mathcal{M}) = v_t(\cdot \mid \mathcal{C}_t)$. Thus, the overall training loss can be expressed as:

$$\mathcal{L}_{\text{total}} = \mathbb{E}_c(\mathcal{L}_{\text{trans}} + \mathcal{L}_{\text{rot}} + \mathcal{L}_{\text{torsion}} + \mathcal{L}_{\text{type}}). \tag{16}$$

At this point, the loss about translation can be expanded as:

$$\mathcal{L}_{\text{trans}} = \mathbb{E}_{\substack{t \sim \mathcal{U}(0,1), p(x_1), \\ p_0(x_0), p_t(x_t \mid x_0, x_1)}} \|v_\theta(x; \mathcal{C}) - x_1^{(i)} + x_0^{(i)}\|_{\mathbb{R}^3}^2. \tag{17}$$

Similarly, the losses for rotation and nucleotide type can be formulated analogously. With these defined, the model is ready for training under specific ligand conditions.

**Inference.** The generation for an RNA of length $N$ is initiated by creating a random point cloud as a structure frame and a random RNA sequence of the same length. Besides, the 3D structure of a ligand is incorporated as the conditional input. This noisy frame and sequence are then iteratively refined into a realistic RNA structure and sequence through the trained model $v_\theta$ and an ODEsolver. The sampling process is detailed in the Algorithm 1.

## 5 Experiments

**Pretraining Dataset.** To establish foundational priors for RNA structural validity, we pre-train RiboFlow on RNAsolo [2], a curated database of single-stranded RNA 3D structures. We filter entries to those with resolution $\leq 4$ Å (as of December 2024) and retain sequences between 30–200 nucleotides to balance structural diversity with computational feasibility, yielding 7,154 high-quality training samples. This length range reflects typical functional RNA motifs while accommodating GPU memory constraints.

**RNA-ligand Interaction Dataset.** We introduce RiboBind, a large standardized dataset for RNA-small molecule interactions, addressing data scarcity in ligand-conditioned design. We discuss dataset construction and perform statistical analysis in the Appendix A.1. We perform *dynamic cropping* and truncate regions distal to the ligand-binding pocket, preserving interaction sites while maximizing data utility (details discussed in Appendix A.3). Meanwhile, we have conducted a comprehensive and objective comparison of existing RNA-ligand datasets, which can be referred to in the Appendix A.2.

**Dataset Splits.** To evaluate model generalization, we employ the following partitioning strategies for RiboBind: (1) *Sequence-based Evaluation:* RNA sequences are clustered at a 50% identity threshold using MMseqs2 [36]. A test set is formed from the cluster centroids, comprising 66 RNA-ligand pairs which include 20 of the most frequently observed ligands. All non-centroid sequences constitute the corresponding training set. (2) *Structure-based Evaluation:* Following the methodology of gRNAde [24], RNA structures are clustered using US-align [49] with a TM-score threshold of 0.45, yielding 277 distinct structural classes. These classes are then partitioned into a training set (249 classes) and a test set (28 classes) at a 9:1 ratio. The final test dataset for this evaluation consists of 28 RNA-ligand pairs, each formed by one representative structure from a held-out test class and its corresponding ligand. (3) *Few-shot Evaluation:* Contains 15 RNA-ligand pairs involving ligands that appear only once in the RiboBind dataset, designed to evaluate low-resource generalization. In the main text, we primarily introduce sequence-based evaluation, while the appendix provides a detailed analysis for structure-based evaluation and few-shot evaluation.

**Evaluation Metrics.** We evaluate our experiment using metrics for both generation quality and binding capability. *For generation quality*, we focus on four metrics: **Validity (val.)**, assessed

by inverse-folding each generated backbone with gRNAde [24] and predicting its structure using RhoFold [35] with 8 generated sequences. The validity is determined by a self-consistency TM-score (scTM) between the predicted and original backbone at the $C4'$ level, with scTM $\geq 0.45$ indicating a valid backbone; **Diversity (div.)**, measured by the proportion of unique structural clusters (identified by qTMclust [49]) among valid samples, Structures with a TM-score $\geq 0.45$ are considered similar, reflecting the structural variability of generated samples; and **Novelty (nov.)**, evaluated using US-align [49] to compare the structure of valid backbones to the training distribution, with higher novelty indicating greater structural divergence. **Sequence Recovery (SR.)**, quantified as the percentage of correctly recovered nucleotides in the co-designed sequence relative to the ground-truth RNA.

*For binding capability*, we assess the complex binding capability *in-silico* by three metrics: **AF-Score (AF.)**, Alphafold3's overall prediction confidence for the complex, incorporating structural confidences (pTM and ipTM), clash penalties, and considerations for disordered regions, where higher scores are better; **GerNAScore** (measured by GerNA-Bind [1]), A recently proposed model for RNA–ligand affinity prediction [45], where higher scores correspond to stronger predicted binding affinity. **VinaScore (vina.)**, the binding free energy (in *kcal/mol*) between RNA and ligand predicted by AutoDock Vina [41], with lower values signifying stronger binding capacity.

**Baselines.** To the best of our knowledge, there is currently no RNA design model specifically targeting small molecules. RNAFlow [32] relies on RoseTTAFoldNA [4] to design RNA structures from predicted protein-RNA complex structures, which cannot predict RNA-ligand complex conformations. MMDiff [31] is also limited to protein-RNA generation. LigandMPNN [11] can only generate proteins for specific ligands. To this end, we choose RNA-FrameFlow [3] for comparison as it is a general RNA structure generation model, despite not being constrained by ligands. For further comparison, we also include a baseline where sequences are randomly generated and folded using RhoFold.

## 5.1 Unconditional RNA Generation

**Setup.** We first evaluate the generative capabilities of each model without conditioning on ligand information. In addition to using the original RNAFrame-Flow model, we assess the effectiveness of the proposed co-design pre-training strategy by comparing: (i) RNA-FrameFlow-R, a retrained version of RNA-FrameFlow using the new pre-training dataset with RNA backbone frames and torsion angle loss; and(ii) RiboFlow, the proposed model trained with the sequence-structure co-design loss. For each model, we set the generation length in the range of [50, 150], *sampling at intervals of 20, and generate 100 samples* for each length to cover the model's sampling space.

**Results.** The experimental results are presented in Table 1, where `step_50` denotes a sampling step of 50. A more detailed comparison of model performance under varying pre-training steps is provided in Table 4. The local structural properties of RNAs generated by RiboFlow and RNAFrame-Flow-R are illustrated in Figure 8. We can draw the following conclusions: (i) The validity of RNAs generated by RiboFlow has shown significant improvement compared to the other baselines. (ii) Due to the addition of sequence feature constraints, the diversity of RNAs generated by the RiboFlow has slightly decreased compared to RNA-FrameFlow-R. (iii) Increasing the sampling steps can enhance the validity of RNA structures to a certain extent. Besides, it can also be noted that the structures generated by RiboFlow are close to the real structures in terms of bond lengths and torsion angles in Figure 8.

Table 1: Unconditional RNA generation.

| METHOD | %VAL.(↑) | DIV.(↑) | NOV.(↓) |
|---|---|---|---|
| RNA-FRAMEFLOW | | | |
| STEP_50 | 25.0 | 0.573 | 0.582 |
| STEP_100 | 30.3 | 0.503 | 0.594 |
| RNA-FRAMEFLOW-R | | | |
| STEP_50 | 26.3 | **0.577** | 0.562 |
| STEP_100 | 30.7 | 0.530 | 0.546 |
| RIBOFLOW | | | |
| STEP_50 | **34.7** | 0.550 | **0.540** |
| STEP_100 | 31.9 | 0.545 | 0.577 |

## 5.2 RNA Generation Guided by Ligand Structure

We evaluate the performance of models conditioned on ligand-bound structures and further examine the impact of incorporating the ground-truth RNA sequence length as a prompt. Hence, we design

---

[1] https://github.com/GENTEL-lab/GerNA-Bind

Table 2: Comparison of RNA generation with sequence-based evaluation.

| | Vina. (↓) | | GerNA. (↑) | %AF. (↑) | %Val.(↑) | Div.(↑) | Nov.(↓) |
|---|---|---|---|---|---|---|---|
| | Top10 | Median | | | | | |
| **Generation Given Ligand Structure and true Length** | | | | | | | |
| Random | -2.76 | -2.55 | 0.101 | 3.11 | - | - | - |
| RNA-FrameFlow | -4.15 | -4.01 | 0.203 | 17.4 | 22.5 | 0.288 | 0.584 |
| RiboFlow | -4.25 | -4.12 | 0.283 | 22.8 | 10.8 | 0.226 | 0.640 |
| + Pre | **-4.39** | **-4.31** | **0.437** | 48.4 | **26.8** | 0.354 | **0.514** |
| + Crop | -4.38 | -4.27 | 0.402 | 34.7 | 12.1 | 0.293 | 0.603 |
| + Pre + Crop | -4.30 | -4.18 | 0.411 | **50.1** | 24.2 | **0.376** | 0.534 |
| **Generation Given Ligand Structure and Sampling Length** | | | | | | | |
| Random | -2.91 | -2.82 | 0.113 | 5.25 | - | - | - |
| RNA-FrameFlow | -4.27 | -4.15 | 0.209 | 22.7 | 22.7 | 0.545 | 0.574 |
| RiboFlow | -4.61 | -4.48 | 0.297 | 25.7 | 10.8 | 0.517 | 0.669 |
| + Pre | **-4.67** | -4.55 | **0.467** | 51.2 | **25.9** | 0.575 | **0.519** |
| + Crop | -4.63 | **-4.56** | 0.456 | 38.9 | 12.7 | 0.522 | 0.613 |
| + Pre + Crop | -4.58 | -4.43 | 0.448 | **54.8** | 25.6 | **0.589** | 0.527 |
| **Generation Given Ligand SMILES and Sampling Length** | | | | | | | |
| Random | -3.32 | -3.13 | 0.107 | 3.54 | - | - | - |
| RNA-FrameFlow | -4.51 | -4.48 | 0.200 | 22.3 | 23.9 | 0.556 | 0.581 |
| RiboFlow | -5.01 | -4.82 | 0.304 | 15.4 | 11.2 | 0.522 | 0.656 |
| + Pre | **-5.12** | **-5.07** | 0.469 | 45.7 | **28.1** | **0.561** | 0.537 |
| + Crop | -5.10 | -4.99 | 0.451 | 27.9 | 12.3 | 0.546 | 0.609 |
| + Pre + Crop | -4.96 | -4.84 | 0.460 | 39.8 | 26.1 | 0.559 | 0.540 |
| RiboFlow-T | -5.07 | -4.94 | **0.472** | **46.2** | 22.4 | 0.550 | **0.524** |

two protocols: (a) using the true RNA sequence length for generation, and (b) employing uniform sampling of RNA lengths within a predefined range. Both protocols are tested on the sequence-based evaluation set. Additionally, we expand our analysis to include the structure-based and few-shot evaluation set (detailed in the Appendix Table 6 and Table 7) to assess the model's performance.

**Setup.** We include three types of model variants in addition to the original model: (i) RiboFlow: The baseline model trained on the RiboBind; (ii) +CROP: Model trained on the RiboBind augmented with dynamic cropping strategy; (iii) +PRE: Model trained on the RiboBind using codesign pretrained weights; (iv) +PRE+CROP: Model trained on RiboBind augmented with dynamic cropping strategy using codesign pretrained weights. The ligand-bound structures are provided to models as guidance. For protocol (a), 100 RNAs of actual length are sampled for each RNA-ligand pair. As for protocol (b), the range is defined as [50,150], sampling at intervals of 20, and generated 100 samples for each length. These candidate RNAs are subsequently evaluated based on their structural validity and binding capability with the ligand.

Table 3: Sequence recovery and structural RMSD comparisons with ground-truth RNA using ligand-bound structures and true sequence lengths on the test set.

| Method | %SR (↑) | RMSD (↓) |
|---|---|---|
| Random | 25.0 | - |
| RNA-FrameFlow | 28.2 | 12.32 |
| RiboFlow | 30.9 | 11.66 |
| +Pre | **33.4** | **9.43** |
| +Crop | 31.2 | 10.32 |
| +Pre+Crop | 31.8 | 11.64 |

**Results.** We first analyze the experimental results provided with the ground-truth RNA lengths, as shown in Table 2 **in the gray-shaded section** and Table 3, **with best results in bold and the second-best results underlined**. Furthermore, we provide detailed experimental results for 24 randomly selected RNA-ligand pairs presented in Figures 12, showcasing the *vina* with actual values as reference. Several interesting conclusions can be summarized: (i) The diversity for all models are significantly lower than those observed in Table 1, which indicates that constraining the RNA length compresses the sequence sampling space, leading to a decrease in diversity; (ii) Models incorporating co-design pre-training strategy (PRE and PRE-CROP) demonstrate a substantial advantage in structural validity and binding capability. Besides, we can also observe that these two models also perform excellently in terms of sequence recovery rate and RMSD metrics in Table 3;

(iii) Compared to the codesign-pretraining, there is no significant improvement in structural validity under the dynamic cropping strategy. This may be because the trimmed RNA fragments may not fold into the real RNA structure, thus the model cannot effectively learn the true structural distribution. However, data augmentation expands the training data, allowing the model to learn a more diverse set of RNA-ligand docking patterns and improve the designed RNA binding capability.

Subsequently, we analyze the experimental results when length is sampling from the pre-defined ranges in Table 2 highlighted **in the blue-shaded section**. Most experimental conclusions are similar to the previous ones, but due to the increase in the length sampling space, models can explore more potential binding patterns, and the results of Vina, GerNA and AFScore increase largely.

We conduct similar experiments on structure-based evaluation and few-shot evaluation, and experimental results can be referred to the Appendix B.5 and Appendix B.6.

## 5.3 RNA Generation Guided by Ligand SMILES

A common challenge in **real-world applications** is the lack of knowledge regarding the precise RNA length and its ligand-bound conformation. This section try to explore whether our model can successfully design small molecule RNA binders despite this significant limitation.

**Setup.** We use RDKit to generate 3D ligand conformers from SMILES, with optimization using the MMFF force field [40]. RNA lengths are sampled at intervals of 20 in the range [50, 150], with 200 structures generated per length. Moreover, we introduce RiboFlow-T, a variant that translates the coordinate system to the ligand centroid during the training of RiboFlow-PRE-CROP. This modification aims to minimize the impact of ligand spatial positioning on RNA-ligand complex modeling. The nucleotide structure frames are subsequently constructed relative to this transformed coordinates.

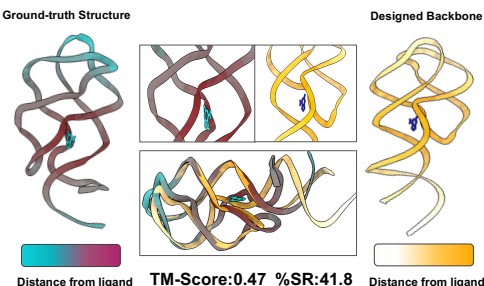

**Results.** The experimental results are shown in Table 2 highlighted **in the pink-shaded section**, which reveals the following insights: RiboFlow-T achieves the highest AFScore and GerNA score, indicating that re-centering on the ligand's centroid during training improves docking pattern reliability. However, this approach introduces inconsistencies with the RNA-centered coordinate system used during pretraining, reducing RNA structure validity. Meanwhile, we observe that when only the SMILES is provided, the predicted RNA-binding affinity is generally higher than that obtained using the true ligand structure. We speculate that this is because the RNA molecule is designed based on the ligand structure generated from SMILES, resulting in a standardized and reproducible input that enhances model robustness. However, the true ligand binding conformation may deviate from the lowest-energy state due to conformational constraints within the complex. In addition, structural variations arising from different sources can introduce input heterogeneity, potentially affecting the model's generalization ability.

Figure 3: *De novo* design of an RNA corresponding to ligand A2F. Left: The ground-truth RNA-ligand complex (PDB ID: 3GOT). Right: RNA backbone designed by RiboFlow. The designed structure achieves a TM-score of 0.47 and a Sequence Recovery (SR) of 41.8% compared to the ground-truth complex.

We also leverage RiboFlow to *de novo* design RNA conditioned on ligand A2F and GNG, with the experimental results presented in the Figure 3 and Appendix B.8.

## 6 Conclusion

In this work, we propose RiboFlow, the first generative model designed for ligand-specific RNA generation. By leveraging RNA backbone frames, torsion angles, and sequence features via conditional flow matching, RiboFlow can effectively capture the conformational flexibility of RNA while improving structural validity through sequence constraints. Additionally, RiboFlow models the 3D RNA-ligand interaction to optimize RNA generation, implicitly enhancing its binding affinity.

Extensive experiments demonstrate the ability of RiboFlow to generate RNA structures with both high validity and target-specific affinity.

## Acknowledgments and Disclosure of Funding

This study has been supported by the National Natural Science Foundation of China [62041209], Natural Science Foundation of Shanghai [24ZR1440600], the Science and Technology Commission of Shanghai Municipality [24510714300], the Lingang Lab Fund [LGL-8888].

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

# APPENDIX

## A  Dataset

### A.1  Dataset Preparation

In this work, we propose RiboBind, the largest RNA-ligand binding dataset to date. RiboBind contains all RNA-small molecule interaction information stored in the RCSB PDB database up to December 2024. The data preparation workflow is described in detail as follows.

(i) *Data collection.* We obtain raw structural data in mmCIF format from the RCSB PDB database using the advanced search query, which supports custom filtering based on specific structural attributes. The search is restricted to structures containing at least one RNA polymer chain and one ligand instance. These criteria are defined using the following query settings: `Structure Attributes -> Number of Distinct RNA Entities >= 1` and `Structure Attributes -> Total Number of Non-polymer Instances >= 1`. This step serves as the initial data collection phase.

(ii) *Ligand filtering.* We adopt a portion of the ligand selection criteria from the HairBoss database [33], while relaxing certain rigid druggability restrictions, aiming to balance ligand diversity and effective identification. Specifically, the following rules are applied during our workflow: (1) the molecular weight of the ligand must fall within the range of 100-1000 Daltons, and (2) the ligand must not be a solvent molecule or a metal ion. Consequently, we expand the original ligand library beyond the 311 ligand classes used in HairBoss by incorporating an additional 237 ligand classes, resulting in a total of 548 unique ligand classes.

(iii) *RNA-ligand interactions determination.* For structures containing valid ligands, RNA-ligand interactions are identified by calculating the minimum distance between any ligand atom and any RNA atom. If the minimum distance is less than or equal to 5 Å, the ligand is considered to interact

with the RNA. Using this criterion, we extract RNA chains and their corresponding interacting ligands from the crystal structures to construct RNA-ligand complexes. Each complex is subsequently decomposed into individual RNA-ligand pairs, where each pair comprises a single RNA chain and its interacting ligand.

(iv) *Redundancy reduction.* To reduce redundancy and improve dataset quality, we employ MM-seqs2 [36] to cluster RNA sequences at a 90% sequence identity threshold. Notably, RNA sequences within the same cluster but bound to different ligands are still treated as distinct RNA-ligand pairs.

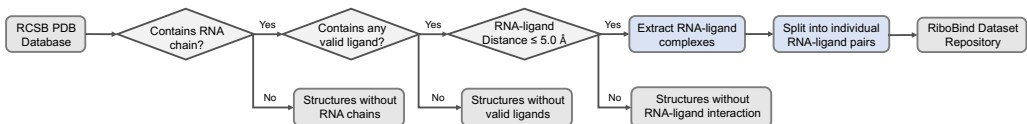

Figure 4: The collection pipeline of RiboBind dataset.

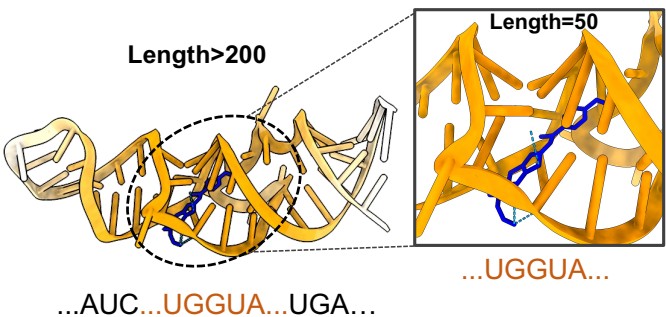

Figure 5: An example of RNA dynamic cropping. Based on a selected nucleotide, we randomly select a cropping length within the range of 30 to 200 to obtain the trimmed RNA-ligand pair for amplifying the existing data.

Through this comprehensive pipeline, we construct the RiboBind dataset, offering a substantially larger and more diverse collection of RNA-small molecule interactions compared to existing datasets. The final RiboBind dataset includes 1,591 RNA-ligand complexes and 3,012 RNA-ligand pairs. In comparison, as of December 2024, the publicly accessible HairBoss dataset comprises only 862 RNA-ligand complexes and 1,471 RNA-ligand pairs. This notable increase in both scale and diversity highlights the value of RiboBind as a robust resource for RNA-small molecule interaction studies. For more detailed statistical information about RiboBind, please refer to Figure 6.

## A.2   Dataset Comparison

Currently, there are also some efforts to collect and curate large-scale RNA-ligand datasets. In addition to the HariBoss dataset referred to in this article, some well-known datasets include PDBBind [30] and RNAmigos2 [8]. PDBBind is a comprehensive database primarily focused on experimentally measured binding affinity data for protein–ligand complexes, though it also includes a small subset of RNA–ligand complexes. In its latest version (v.2024), it contains 234 RNA–ligand complexes. According to our inspection, all of these entries are already included in the RiboBind dataset, making PDBBind essentially a high-quality subset of RiboBind in the context of RNA–ligand interactions.

The RNAmigos2 dataset, on the other hand, was introduced alongside the recently proposed RNA structure-based virtual screening tool RNAmigos2. This dataset includes 1,740 experimentally validated RNA–ligand binding site structures. After deduplication, we verified that all of these experimental structures are also present in the RiboBind dataset. In addition to the experimental data, the RNAmigos2 dataset also contains 1.3 million synthetic affinity data points, generated via in silico molecular docking. These synthetic interactions were computed by docking approximately 800 ligands (sourced from the ChEMBL database) against the 1,740 experimentally solved RNA structures

in all pairwise combinations. Since RiboBind currently only includes experimentally determined structural data, it remains fully consistent with the experimental portion of the RNAmigos2 dataset. However, we plan to incorporate the synthetic data in future versions of RiboBind to increase the chemical diversity and overall scale of the dataset, thereby enhancing the generative performance of our models.

### A.3 Data Augmentation

To prevent graphics memory overflow during experiments, the RNA length is restricted to the range of 30-200 nucleotides during training. However, the original RiboBind dataset contains a limited number of samples within this length range, which is insufficient to satisfy the large-scale data requirements of deep generative models. To address this challenge, we introduce a data augmentation strategy called *dynamic cropping*, designed to fully leverage existing data resources by modifying RNA sequences longer than 200 nucleotides.

In order to ensure that the cropped RNA sequences retain interactions with small molecules, the cropping strategy is informed by interaction data from existing datasets. Specifically, we select RNA bases involved in interactions with the small molecules as candidate bases. By calculating the total interaction counts between each base and every atom in the small molecules, we can identify the top three bases with the highest interaction frequencies as cropping centers. The cropping length is randomly sampled within the range of 30–200 nucleotides. Using the selected base as the center, the RNA sequence is cropped symmetrically, with half the total length taken from both upstream and downstream of the base, forming the final RNA-ligand pair. For more detailed statistical information about the amplified RiboBind dataset, please refer to Figure 7. This approach ensures that the cropped RNA sequences retain critical interaction information for downstream tasks.

Through this dynamic cropping strategy, the number of qualified RNA-ligand pairs in the original RiboBind dataset increased from 1,061 to 4,445, significantly expanding the available training data.

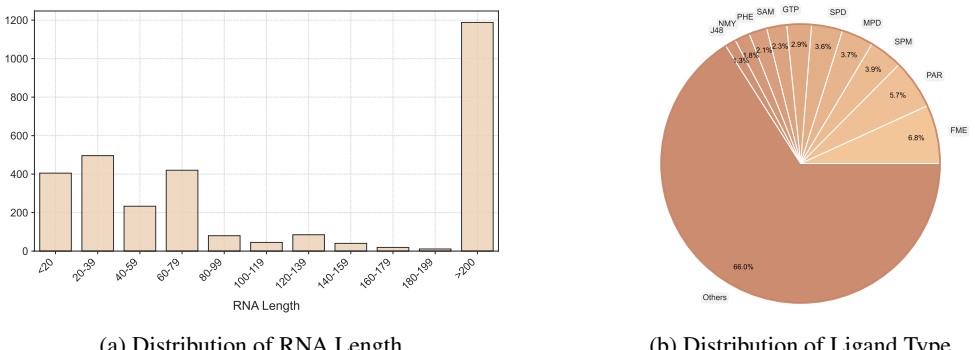

(a) Distribution of RNA Length          (b) Distribution of Ligand Type

Figure 6: Statistics of the originally collected RiboBind dataset. (a) The number of RNAs within different length ranges, and the vast majority of RNAs exceed the 200-nucleotide limit; (b) The proportion of the top ten categories of ligands.

## B  More Results and Analysis

### B.1 The Impact of Pre-training Steps on RNA Generation

To fully demonstrate the impact of different pre-training steps on RNA generation, we comprehensively show the changes in model performance under various pre-training steps. We additionally introduced the **scRMSD** metric to evaluate the performance changes. scRMSD refers to self-consistency Root Mean Squared Deviation between the generated and the predicted backbone atoms to reflect structural validity. During the pre-training stage, a batch size of 32 is selected, utilizing 4 A100 80GB accelerator cards, completing 200K pre-training steps in nearly 20 hours. The experimental results are shown in Table 4. 50K_step_50 means that the model samples with a sampling step of 50 after training for 50K steps.

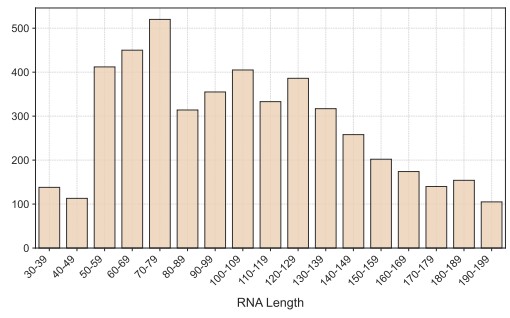

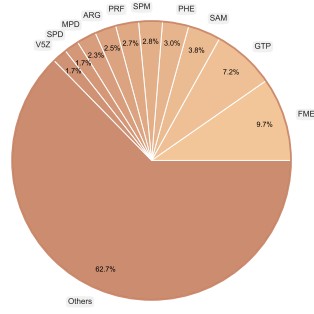

(a) Distribution of RNA Length  (b) Distribution of Ligand Type

Figure 7: Statistical analysis of the RiboBind dataset, amplified using a *dynamic cropping* strategy after removing sequences shorter than 30 nucleotides, reveals the following: (a) The amplified dataset exhibits a more balanced distribution of RNA lengths compared to the original RiboBind dataset, making it better suited for model training. (b) The top ten ligand categories in the amplified dataset are still predominantly composed of the seven major ligands (FME, GTP, SAM, PHE, SPM, MPD, and SPD) found in the original dataset. This suggests that the ligand bias introduced by the amplification process remains within an acceptable range.

Table 4: Detailed results of unconditional generation under different pre-training steps.

| METHOD | %scTM(↑) | %scRSMD(↑) | DIVERSITY(↑) | NOVELTY(↓) |
|---|---|---|---|---|
| **RNA-FRAMEFLOW** | | | | |
| STEP_50 | 25.0 | 24.7 | **0.573** | **0.582** |
| STEP_100 | **30.3** | **28.3** | 0.503 | 0.594 |
| **RNA-FRAMEFLOW-R** | | | | |
| 50K_STEP_50 | 14.3 | 16.0 | **0.733** | 0.677 |
| 50K_STEP_100 | 20.3 | 19.3 | 0.703 | 0.670 |
| 100K_STEP_50 | 27.7 | 26.7 | 0.550 | 0.597 |
| 100K_STEP_100 | 28.3 | **27.0** | 0.547 | 0.586 |
| 150K_STEP_50 | 26.3 | 22.7 | 0.577 | 0.562 |
| 150K_STEP_100 | **30.7** | 24.8 | 0.530 | **0.546** |
| 200K_STEP_50 | 25.0 | 22.3 | 0.590 | 0.553 |
| 200K_STEP_100 | 27.0 | 24.3 | 0.537 | 0.559 |
| **RIBOFLOW** | | | | |
| 50K_STEP_50 | 14.0 | 21.7 | **0.710** | 0.685 |
| 50K_STEP_100 | 20.7 | 23.3 | 0.643 | 0.511 |
| 100K_STEP_50 | 30.3 | 29.0 | 0.583 | 0.589 |
| 100K_STEP_100 | 31.0 | 27.3 | 0.620 | 0.574 |
| 150K_STEP_50 | **34.7** | **31.3** | 0.550 | **0.540** |
| 150K_STEP_100 | 31.9 | 30.7 | 0.545 | 0.577 |
| 200K_STEP_50 | 32.7 | 26.7 | 0.520 | 0.581 |
| 200K_STEP_100 | 30.7 | 27.0 | 0.493 | 0.606 |

## B.2 The Local Structure of RNA Analysis

To demonstrate the validity of the generated RNA structures, we conduct a detailed comparison between the RNA structures generated by RNAFrame-Flow-R and RiboFlow within the [50,150] range and the actual RNA structure distributions. We primarily illustrate histograms of the probability distributions for the bond distances between nucleotides, the bond angles between nucleotide triplets, and the torsion angles, as shown in Figure 8. It can be observed that the structures of the RNA we generated are similar in distribution to those of real RNA, capable of reproducing the characteristics of these local structures.

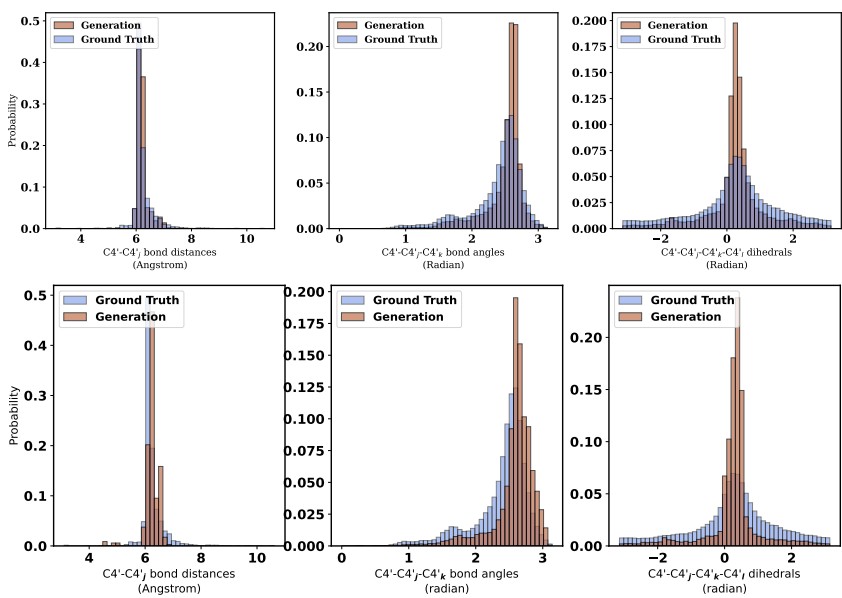

Figure 8: Comparison of probability distribution histograms for nucleotide bond distances, bond angles, and torsion angles between generated and real RNA structures. Top: RNA-FrameFlow-R; Bottom: RiboFlow. RiboFlow shows a closer match to the real data distribution, particularly in bond angles, indicating improved geometric realism over RNA-FrameFlow-R.

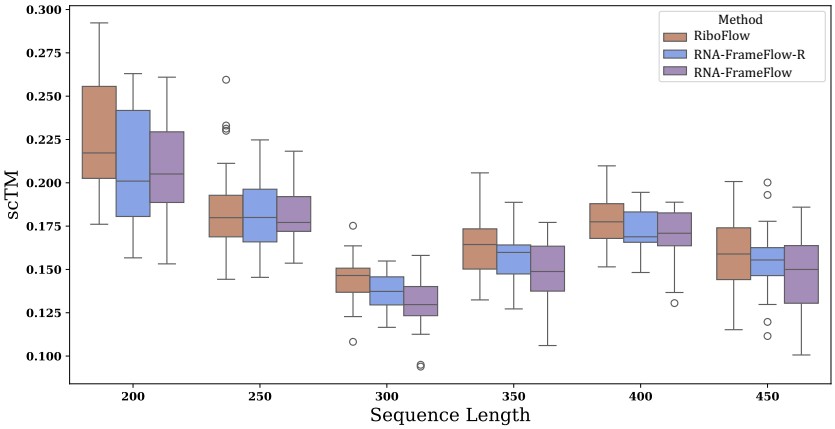

Figure 9: The impact of sampling lengths on model generation quality.

## B.3   The Impact of Sampling Lengths on Model Generation Quality

Due to the pre-training data distribution limitations in RNASolo (which is primarily concentrated in the range of 30–200), we set the RNA sampling range to [50, 150]. However, we are willing to explore RiboFlow's performance in designing longer RNA sequences. To this end, we conducted additional experiments, performing unconditional RNA backbone design for sequences of lengths [200, 250, 300, 350, 400, 450]. Notably, attempting longer sequences on an RTX 4090 GPU (24GB memory) triggers an OOM error due to the memory requirements of RhoFold. The scTM experimental results for each length are as shown in Figure 9.

Our findings indicate that for longer sequences, RNA validity declines significantly compared to the performance in the [50, 150] range reported in the paper. However, it is important to note that RhoFold explicitly states that its training data only includes nucleotide sequences of lengths 16–256, and its accuracy beyond this range has not been fully validated. Therefore, this experiment provides

Table 5: Structure generation validity of the model under unconditional generation, evaluated using different backbone atoms ($C3'$, $C4'$, $C5'$) and their average.

| METHOD | %scRMSD(Avg.) | %scRMSD($C3'$) | %scRMSD($C4'$) | %scRMSD($C5'$) | %scTM(Avg.) | %scTM($C3'$) | %scTM($C4'$) | %scTM($C5'$) |
|---|---|---|---|---|---|---|---|---|
| **RNA-FRAMEFLOW** | | | | | | | | |
| SAMPLE_STEP_50 | 27.0 | 27.7 | 27.0 | 27.0 | 26.3 | 27.0 | 26.7 | 26.7 |
| SAMPLE_STEP_100 | 29.3 | 29.7 | 30.3 | 29.7 | 31.6 | 32.0 | 31.3 | 30.6 |
| **RNA-FRAMEFLOW-R** | | | | | | | | |
| SAMPLE_STEP_50 | 25.7 | 25.7 | 25.0 | 25.7 | 26.7 | 26.0 | 26.0 | 25.7 |
| SAMPLE_STEP_100 | 26.3 | 26.3 | 26.0 | 26.7 | 30.0 | 31.3 | 30.0 | 31.6 |
| **RIBOFLOW-CODESIGN** | | | | | | | | |
| SAMPLE_STEP_50 | 32.7 | 32.3 | 32.7 | 32.3 | 36.7 | 38.3 | 37.0 | 36.3 |
| SAMPLE_STEP_100 | 31.3 | 31.0 | 31.3 | 31.3 | 33.7 | 35.7 | 33.3 | 31.3 |

only a rough assessment of the model's generalization ability. We look forward to advancements in RNA engineering, where more powerful structural prediction models will enable more complex RNA designs.

### B.4 Using Different Backbone Atoms for Model Evaluation

In the paper, we used the $C4'$ atom to evaluate model performance in experimental results. To assess the validity of generated RNA structures, we employed scTM and scRMSD metrics. The TM-score was calculated using US-align [49], which by default selects a single backbone atom for structural evaluation.

Recognizing that a multi-atom approach could provide a more comprehensive assessment, we expand the evaluation to include the backbone atoms available in US-align ($C3'$, $C4'$, $C5'$, see `https://zhanggroup.org/US-align/` for detailed information). We computed scTM and scRMSD for each individual atom and their mean values to enhance evaluation robustness. Note that scRMSD refers to the RMSD between the structure predicted by RhoFold and the structure generated by the model. We refer to the settings in RNA-FrameFlow [3] and define samples with scRMSD below 4.3 Å as valid structures. We report the performance of the model using different backbone atoms under unconditional generation tasks.

The results can be found in Table 5. Please note that each column in the figure represents the validity of the structure generated by using different metrics for calculation. For example, %scRMSD(Avg.) represents the proportion of valid structures generated by calculating scRMSD using the average metric of these three atoms $C3'$, $C4'$, and $C5'$ simultaneously. It can be indicate that evaluations based on multiple atoms yield consistent performance trends, supporting a more accurate reflection of the model's structural generation quality.

### B.5 Experimental Results on Structure-based Evaluation

Following the experimental setup of Section 5.2, we conduct similar experiments on the structure-based evaluation set, with the experimental results as shown in Table 6. Our findings show that RiboFlow and its variants still exhibit strong performance under structural-based division.

### B.6 Experimental Results on Few-shot Evaluation

Following the experimental setup of Section 5.2, we conduct similar experiments on the few-shot set, with the experimental results as shown in Table 7. It can be observed that despite the model being exposed to only a small amount of sample information during training, it still demonstrated excellent performance. In addition, we have also presented the detailed results of vina scores for these fifteen RNA-ligand pairs in the Figure 13, fully demonstrating the experimental performance of different model variants.

### B.7 Sampling Time Efficiency Analysis

In this section, we evaluate the sampling efficiency of RiboFlow in comparison to RNA-FrameFlow, a critical factor for the practical utility of generative models. This analysis measures the time required to generate RNA structures of varying lengths (approximately 50 to 300 nucleotides) using both 50

Table 6: Comparison of RNA generation by structure-based evaluation. The best results are highlighted in bold, while the second-best results are underlined. The table presents two experimental results: one based on a fixed RNA length and ligand-bound structure, and another based on ligand-bound structures with RNA length sampling. Colored boxes denote different experimental setups.

| | VINA. (↓) | | GERNA. (↑) | %AF. (↑) |
|---|---|---|---|---|
| | TOP10 | MEDIAN | | |
| GENERATION GIVEN LIGAND STRUCTURE AND TRUE LENGTH | | | | |
| RANDOM | -2.98 | -2.85 | 0.121 | 5.32 |
| RNA-FRAMEFLOW | -4.33 | -4.25 | 0.225 | 23.9 |
| RIBOFLOW | -5.07 | -5.01 | 0.231 | 25.2 |
| + PRE | -5.32 | -5.26 | **0.452** | **52.3** |
| + CROP | -5.17 | -5.09 | 0.389 | 40.2 |
| + PRE + CROP | **-5.43** | **-5.39** | 0.439 | 49.7 |
| GENERATION WITH GIVEN RNA LENGTH AND LIGAND CONFORMATION | | | | |
| RANDOM | -2.32 | -2.15 | 0.110 | 6.47 |
| RNA-FRAMEFLOW | -4.35 | -4.24 | 0.217 | 17.4 |
| RIBOFLOW | -5.36 | -5.21 | 0.245 | 23.1 |
| + PRE | **-5.64** | **-5.49** | **0.437** | **45.3** |
| + CROP | -5.41 | -5.30 | 0.402 | 33.5 |
| + PRE + CROP | -5.56 | -5.38 | 0.393 | 41.3 |

Table 7: Comparison of model performance on the few-shot evaluation set. The best results are highlighted in bold, while the second-best results are underlined. The table presents two experimental results: one based on a fixed RNA length and ligand-bound structure, and another based on ligand-bound structures with RNA length sampling. Colored boxes denote different experimental setups.

| | VINA. (↓) | | GERNA. (↑) | AF. (↑) |
|---|---|---|---|---|
| | TOP10 | MEDIAN | | |
| GENERATION WITH GIVEN RNA LENGTH AND LIGAND CONFORMATION | | | | |
| RANDOM | -2.16 | -2.03 | 0.125 | 7.11 |
| RNA-FRAMEFLOW | -3.95 | -3.88 | 0.206 | 16.4 |
| RIBOFLOW | -4.63 | -4.49 | 0.276 | 22.8 |
| + PRE | **-4.74** | **-4.63** | **0.415** | 47.1 |
| + CROP | -4.56 | -4.51 | 0.401 | 25.6 |
| + PRE + CROP | -4.47 | -4.41 | 0.376 | **49.9** |
| GENERATION WITH GIVEN RNA LENGTH AND LIGAND CONFORMATION | | | | |
| RANDOM | -2.23 | -2.15 | 0.110 | 8.84 |
| RNA-FRAMEFLOW | -4.35 | -4.01 | 0.207 | 19.1 |
| RIBOFLOW | -4.87 | -4.66 | 0.287 | 24.1 |
| + PRE | **-5.04** | **-4.89** | **0.433** | 48.6 |
| + CROP | -4.91 | -4.75 | 0.412 | 32.4 |
| + PRE + CROP | -4.96 | -4.82 | 0.403 | **51.3** |

and 100 sampling steps. All experiments were conducted on an identical hardware platform to ensure a fair and consistent comparison of computational overhead.

Figure 10 illustrates the sampling time costs. At 50 sampling steps, RiboFlow exhibits a sampling time that is largely comparable to RNA-FrameFlow across the tested sequence lengths, with RiboFlow's time being slightly higher in most instances. When the number of sampling steps is increased to 100, RiboFlow incurs a more noticeable increase in computational time relative to RNA-FrameFlow. This difference in sampling time between the two models becomes more pronounced with increasing sequence length at 100 steps, indicating that the processing of additional conditioning information in RiboFlow contributes to a steeper scaling of its sampling time, particularly for longer sequences.

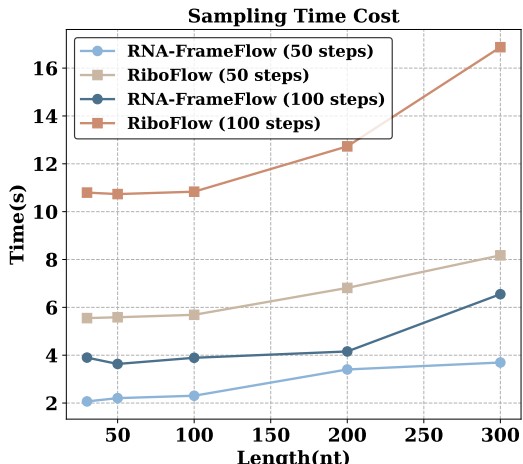

Figure 10: Sampling time comparison between RiboFlow and RNA-FrameFlow. Time in seconds is plotted against RNA sequence length (nt) for both 50 and 100 sampling steps.

Nevertheless, the sampling times for RiboFlow remain within a practical and acceptable range for typical applications.

## B.8 Using RiboFlow for *De Novo* RNA Design

In this section, we aim to validate RiboFlow's capability in designing *de novo* high-affinity RNAs for specific ligand targets. As a case study, we explore two design scenarios: one where the actual RNA length and small molecule binding structure (ligand A2F[2]) are provided, and another where neither the RNA length nor the binding structure (ligand GNG[3]) is given.

**Pipeline.** *For ligand A2F.* We employ RiboFlow to generate RNA sequences of length 67, which are then docked against A2F ligand-bound conformers using AutoDock Vina. A specific RNA candidate is selected based on a composite score integrating Vina scores and ligand pose validity. The corresponding ground-truth RNA sequence and the designed RNA sequence are presented in the results.

*For ligand GNG.* We first leverage RiboFlow to generate RNAs ranging from 50 to 150 nucleotides in length. These RNA sequences are then docked against GNG conformers, generated by RDKit, using AutoDock Vina. A 90-nucleotide RNA candidate is selected based on a composite score that integrates vina and ligand pose validity. To validate this candidate, gRNAde is applied to determines its corresponding sequence, and AlphaFold3 predicts the structure of the resulting RNA-GNG complex.

**Case Analysis.** *For ligand A2F.* The ground-truth RNA-ligand complex used in our study has a PDB ID of 3GOT. The TM-score between our designed RNA and the native RNA is 0.47, with a sequence recovery rate of 41.8%, as shown in Figure 3. Below, we provide both the designed RNA sequence and the corresponding native RNA sequence for comparison:

```
>Ground-truth RNA Sequence (3GOT)
GGACAUAUAAUCGCGUGGAUAUGGCACGCAAGUUUCUACCGGGCACCGUAAAUGUCCGAUUAUGUCC
>Designed RNA Sequence
CGGUGGGAAGGGGUGAGGCCAGGCUAUACCUGGCGCAACGUCUCACCUUUAAUGGGCAAGCCUUGCC
```

*For ligand GNG.* The AlphaFold3-predicted complex structure exhibits a TM-score of 0.41 compared to the RiboFlow-designed RNA backbone, demonstrating the validity of the designed RNA. Furthermore, as illustrated in Figure 11, a high consistency is observed between the GNINA-predicted binding pocket and the AlphaFold3-predicted binding region. These findings provide strong support for RiboFlow's potential to facilitate the design of RNAs with high binding affinity.

---

[2]https://www.rcsb.org/ligand/A2F
[3]https://www.rcsb.org/ligand/GNG

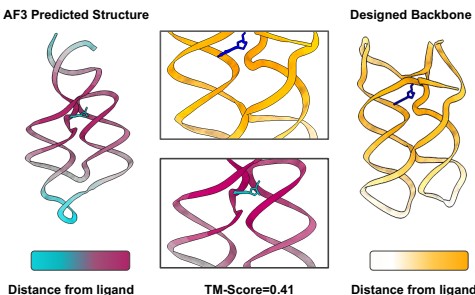

Figure 11: *De novo* design of a 90-nucleotide RNA corresponding to the ligand GNG with a molecular weight of 267.2 Da. The RNA-ligand complex structure is predicted by AlphaFold3, whereas another is generated by RiboFlow and docking using GNINA. The AlphaFold3-predicted complex structure exhibits a TM-score of 0.41 compared to the RiboFlow-designed RNA backbone, demonstrating the validity of the designed RNA.

# C    More Details of RiboFlow Training and Generation

## C.1    Model Training Details

RiboFlow utilizes several hyperparameters in the experiments, which are crucial for the model's training and sampling processes. Therefore, we provide some key hyperparameters to facilitate the reproduction of our experiments in Table 8. The optimal hyperparameters are indicated in bold.

Table 8: Hyperparameters for the RiboFlow.

| Category | Hyperparameter | Value |
| --- | --- | --- |
| IPA Module | Atom embedding dimension | 256 |
| | Hidden dimension | 16 |
| | Number of blocks | 8 |
| | Query and key points | 8 |
| | Number of heads | 8 |
| | Key points | 12 |
| Transformer | Number of heads | 4 |
| | Number of layers | 4 |
| Torsion MLP | Input dimension | 256 |
| | Hidden dimension | 128 |
| Ligand Module | Number of atom type | 95 |
| | Number of RBF | 16 |
| | Distance range | [0.05, 6.0] |
| Training Schedule | Translations (training / sampling) | linear / linear |
| | Rotations (training / sampling) | linear / exponential |
| | Number of sampling steps | [50, 100] |
| | Optimizer | AdamW |
| | Learning rate | 1e-4 |
| | Number of GPUs | 4 |
| | Batch size | [4, 8, 16, 32] |

---

**Algorithm 1** RiboFlow: Inference

---

1: **Input:** Ligand structure $\mathcal{M}$, Sampling steps $T$, Initial RNA structure $\mathcal{T}_0$ with length $N$, Trained model $v_\theta$
2: Initialize: $steps \leftarrow 0, t \leftarrow 0, \Delta t \leftarrow 1/T$
3: Initialize: complex $\mathcal{C}_0$ with RNA $\mathcal{T}_0$ and ligand $\mathcal{M}$
4: **while** $steps < T$ **do**
5: $\quad x_{t+\Delta t}^{(i)} \leftarrow x_t^{(i)} + \boldsymbol{v}_\theta(x_t^{(i)}, t; \mathcal{C}_t)\Delta t$
6: $\quad r_{t+\Delta t}^{(i)} \leftarrow r_t^{(i)} \exp\left(\boldsymbol{v}_\theta(r_t^{(i)}, t; \mathcal{C}_t)\Delta t\right)$
7: $\quad s_{t+\Delta t}^{(i)} \leftarrow \text{norm}\left(s_t^{(i)} + \boldsymbol{v}_\theta(s_t^{(i)}, t; \mathcal{C}_t)\Delta t\right)$
8: $\quad$ Sample nucleotide type: $c_{t+\Delta t}^{(i)} \sim s_{t+\Delta t}^{(i)}$
9: $\quad t \leftarrow t + \Delta t, steps \leftarrow steps + 1$
10: **end while**
11: Calculate: $\hat{\Phi}_1^{(i)} \leftarrow \boldsymbol{v}_\phi(\hat{x}_1^{(i)}, \hat{r}_1^{(i)})$
12: **Return:** Final complex $\mathcal{C}_1$

---

## C.2 Model Inference

We provide the pseudocode in Algorithm 1 for RiboFlow inference to intuitively demonstrate the sampling process of the model.

# D Flow Matching

Flow Matching (FM) [28] is a method designed to reduce the simulation required for learning Continuous Normalizing Flows (CNFs), which is a class of deep generative models that generate data by integrating an ordinary differential equation (ODE) over a learned vector field. In this section, we will provide a concise overview of the flow matching approach.

A continuous normalizing flow $\phi_t(\cdot) : \mathcal{M} \to \mathcal{M}$ on a manifold $\mathcal{M}$ is defined as the solution to a time-dependent vector field $v_t(x) \in \mathcal{T}x\mathcal{M}$, where $\mathcal{T}x\mathcal{M}$ represents the tangent space of $\mathcal{M}$ at $x \in \mathcal{M}$:

$$\frac{d}{dt}\phi_t(x) = v_t(\phi_t(x)), \quad \phi_0(x) = x. \tag{18}$$

The parameter $t$ evolves within $[0, 1]$, and the flow transforms a simple prior density $p_0$ into the data distribution $p_1$ via the push-forward equation $p_t = [\phi_t]_* p_0$. The density of $p_t$ is given by:

$$p_t(x) = [\phi_t]_* p_0(x) = p_0(\phi_t^{-1}(x)) e^{-\int_0^t \text{div}(v_t)(x_s)\,ds}. \tag{19}$$

The sequence of distributions $p_t : t \in [0, 1]$ is referred to as the *probability path*. While the vector field $v_t$ that generates a specific $p_t$ is generally intractable, it can be approximated efficiently by expressing the target probability path as a mixture of simpler *conditional* probability paths, $p_t(x|x_1)$. These conditional paths satisfy $p_0(x|x_1) = p_0(x)$ and $p_1(x|x_1) \approx \delta(x - x_1)$. The unconditional probability path $p_t$ can then be recovered as the average of the conditional paths with respect to the data distribution: $p_t(x) = \int p_t(x|x_1) p_1(x_1), dx_1$.

To describe this further, let $u_t(x|x_1) \in \mathcal{T}x\mathcal{M}$ denote the *conditional vector field* generating the conditional probability path $p_t(x|x_1)$. Flow matching builds on the insight that the unconditional vector field $v_t$ can be learned by aligning it with the conditional vector field $u_t(x|x_1)$ using the following objective:

$$\mathcal{L}_{\text{CFM}} := \mathbb{E}_{t, p_1(x_1), p_t(x|x_1)}\left[\|v_t(x) - u_t(x|x_1)\|_g^2\right], \tag{20}$$

where $t \sim \mathcal{U}([0, 1])$, $x_1 \sim p_1(x_1)$, $x \sim p_t(x|x_1)$, and $\|\cdot\|_g^2$ represents the norm induced by the Riemannian metric $g$.

The objective can be reparameterized through the conditional flow, $x_t = \psi_t(x_0|x_1)$, where $\psi_t$ satisfies $\frac{d}{dt}\psi_t(x) = u_t(\psi_t(x_0|x_1)|x_1)$ with initial condition $\psi_0(x_0|x_1) = x_0$. This allows the conditional flow matching loss to be reformulated as:

$$\mathcal{L}_{\text{CFM}} = \mathbb{E}_{t, p_1(x_1), p_0(x_0)}\left[\|v_t(x_t) - \dot{x}_t\|_g^2\right]. \tag{21}$$

Once trained, samples can be generated by simulating Equation (18) using the learned vector field $v_t$.

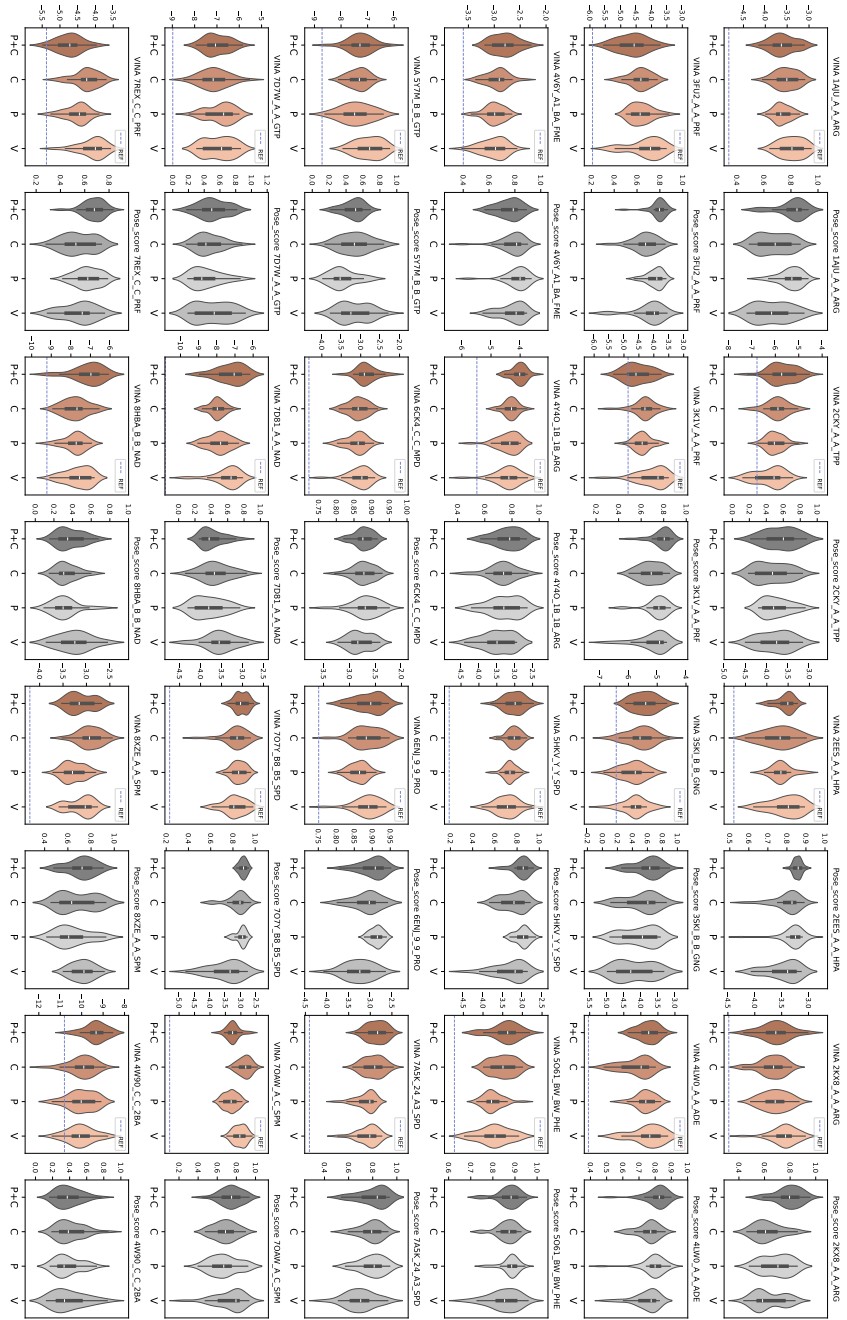

Figure 12: The results of vina score for the 24 RNA-ligand pairs in the sequence-based evaluation set. We additionally provide the experimental values under real conditions, indicated by a blue dashed line.

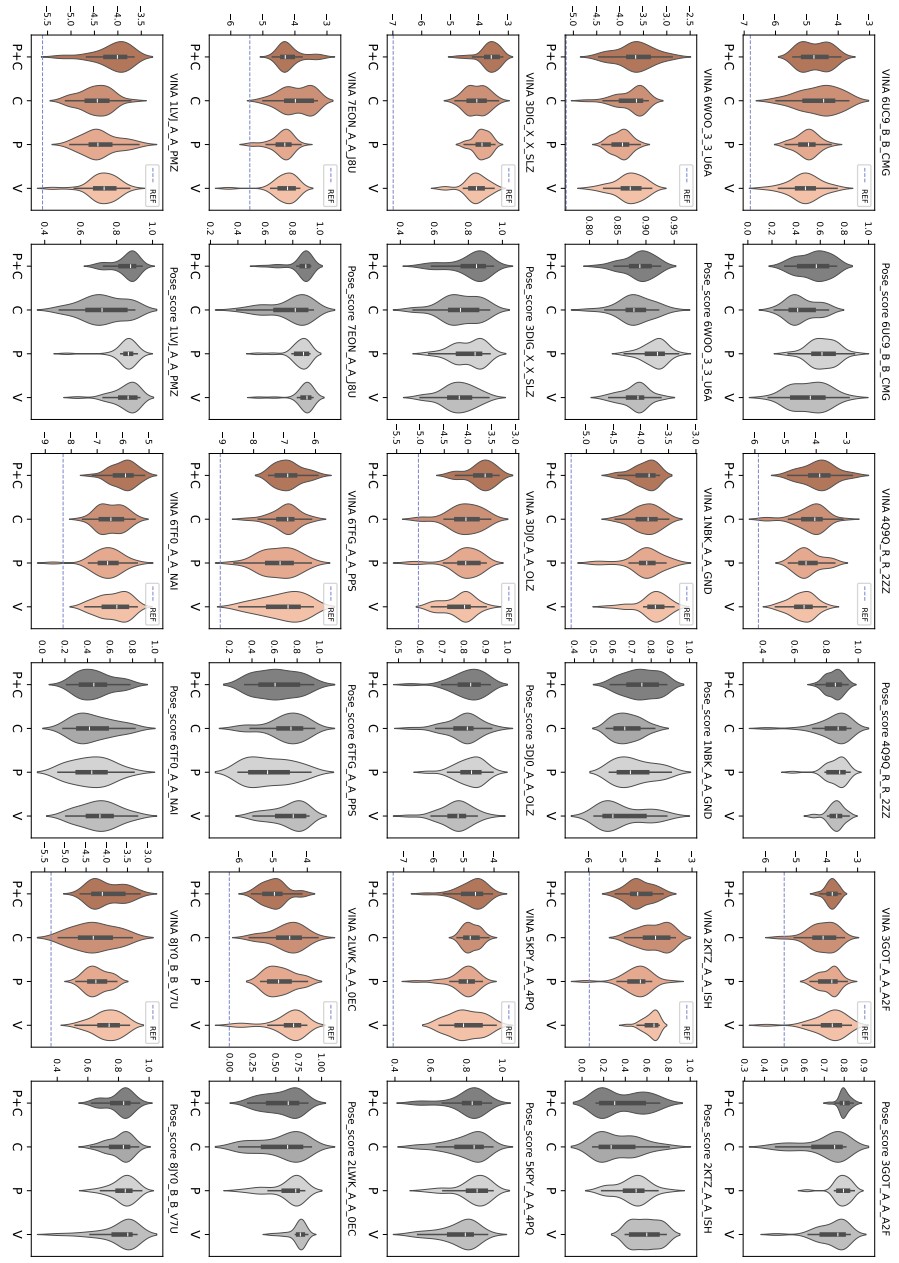

Figure 13: The results of vina for the 15 RNA-ligand pairs in the few-shot set. We additionally provide the experimental values under real conditions, indicated by a blue dashed line.

# E   Limitations and Broader Impact

RiboFlow has certain limitations, which we briefly discuss here. The model's pre-training heavily depends on RNASolo, a dataset of RNA structures predominantly spanning lengths of 30–200 nucleotides. Consequently, RiboFlow's generative performance on longer RNA sequences remains constrained. We anticipate, however, that with continued advances in biotechnology, more high-quality RNA crystal structures will become available, which will help alleviate this limitation. In addition, our current evaluations of RNA-ligand binding affinities are entirely computational and have yet to be validated experimentally. To address this, we are actively collaborating with experimental partners to synthesize the designed RNAs and assess their binding affinities through wet-lab experiments. We are optimistic that these future experimental results will provide stronger empirical support for our approach.

As for broader impacts, our work has many potential application areas. For example, designing fluorescent aptamers that bind to specific molecules as biosensors to detect contamination, or targeting specific biological metabolites for disease treatment. Under the premise of sound legal regulation, we believe that RiboFlow will play a significant role in RNA engineering in the future to benefit all of humanity.

