# OpenReview forum: "RiboFlow: Conditional De Novo RNA Co-Design via Synergistic Flow Matching"
_NeurIPS.cc/2025/Conference — NeurIPS 2025 poster_

### Official Review · Reviewer_YV5K · 2025-06-21

**Clarity:** 2
**Significance:** 2
**Originality:** 2
**Rating:** 4
**Confidence:** 3

**Summary:**

This paper presents an approach with the introduction of RiboFlow, a synergistic flow matching model designed to co-design RNA structures and sequences conditioned on target molecules. The authors curate RiboBind, a comprehensive dataset derived from the PDB database, which stands as the largest RNA-ligand interaction dataset to date. Experimental results are promising, demonstrating that RiboFlow achieves superior validity in unconditional RNA generation when compared to RNA-FrameFlow, while maintaining comparable performance across other key metrics. The following experiments validate some of the design choices.

**Questions:**

1. I was wondering if you could clarify the meaning of $ p_t $ in equation (1) within the context of RNA design. While the mathematical formulation is provided, a brief explanation of its role in the RNA-specific application would be helpful for better understanding.
2. I'm interested in understanding the scale of the RiboBind dataset. With 1,591 RNA-ligand complexes and 3,012 RNA-ligand pairs, how does this compare to other existing datasets in the field? Is this considered large-scale and comprehensive for RNA-ligand interaction studies?
3. Looking at Table 2, I see that the pre-training strategy appears to yield the best results among the three strategies presented. Could you explain why the dynamic cropping strategy is still important or how it contributes to the overall effectiveness of the model, even if it doesn't outperform pre-training in most metrics?

**Ethical Concerns:**

["NO or VERY MINOR ethics concerns only"]

**Final Justification:**

All of my concerns have been addressed. This paper should be accepted, in my opinion.

**Limitations:**

1. I am impressed by the approach of RiboFlow in tackling the challenges of ligand-conditioned RNA generation, particularly its ability to model RNA’s dynamic conformations and achieve high binding affinity. Given the flexibility of the flow-matching framework, I was wondering if there are plans to explore its application to other types of conditioned RNA generation, such as protein-conditioned RNA design or unconditioned RNA generation, to broaden its scope as a versatile tool for RNA design?

**Paper Formatting Concerns:**

To enhance the manuscript's polish, I kindly suggest reviewing the citation formatting for consistency. For instance, I noticed that RNAFlow, which was accepted by ICML 2024, is cited using its arXiv version. Updating this to reflect the official ICML 2024 publication would ensure accuracy and alignment with standard citation practices.

**Quality:**

3

**Strengths And Weaknesses:**

Strengths
1. This paper proposes a novel idea by incorporating RNA backbone, structure and sequence features into a unified flow-based framework. The results are promising, particularly for ligand-conditioned RNA generation.

Weaknesses
1. The technical contribution seems incremental. The proposed method builds upon established works such as RNA-FrameFlow[1] and DFM[2], it would be beneficial for the authors to more clearly articulate the unique technical contributions of this work. For instance, the formulation of RNA frames and structure modeling in SE(3) space are derived from RNA-FrameFlow, and the mapping of discrete sequence tokens to continuous flow from DFM, besides the integration of these elements, are there any unique technical contribution of the this work ? Please introduce them in details.
2. The fundamental design choice is not very well validated. The choice of a flow-based method for RNA design, while effective, could benefit from a more thorough justification, especially given existing methods like RNAFlow[3]. It would be helpful if the authors could elaborate on why a new flow-based framework is necessary and how it addresses limitations in current approaches. Additionally, since AlphaFold3 supports RNA-ligand interaction prediction and is used in this work's evaluation, it might be worth discussing why not replace RosettaFold2NA with AlphaFold3, which could potentially extend RNAFlow to RNA-ligand cases.
3. The presentation and structure of the paper could be improved for clarity. While the authors claim three technical contributions—synergistic flow-matching, the RiboBind dataset, and a multi-faceted evaluation pipeline—the distribution of content seems uneven. The model design sections, such as Preliminary and 4.2, are detailed but largely based on previous works[1][2]. In contrast, the dataset (RiboBind) and evaluation protocol receive less explanation, which might underrepresent their significance. It would be valuable to see more novelty in the dataset curation and evaluation design. Additionally, some technical claims, such as “RiboFlow explicitly models RNA’s dynamic conformations while enforcing sequence-structure consistency to improve validity,” could be better supported by the model design description. For instance, in section 4.4, the focus is on modeling discrete nucleotides into flow, but it is not clear how sequence-structure consistency is explicitly enforced. Furthermore, the role of section 4.5 is somewhat ambiguous; it would be helpful to understand its integration into the overall pipeline—whether it generates ligand-guided RNA backbone inputs or refines post-prediction structures. Clarifying these aspects would strengthen the paper’s coherence and impact.

In light of the points discussed, I believe the paper would benefit from further revision to address the concerns raised, particularly in terms of clarifying the unique contributions, justifying the methodological choices, and improving the overall structure and presentation. While the current version may not fully meet the standards for a NeurIPS publication, the work shows promise and, with the suggested improvements, could be a strong candidate for future conferences. I encourage the authors to carefully consider these suggestions and continue their efforts in this area of research, which has potential applications in therapeutics and synthetic biology.

[1] Flow Matching for de novo 3D RNA Backbone Design.

[2] Generative Flows on Discrete State-Spaces: Enabling Multimodal Flows with Applications to Protein Co-Design. ICML 2024

[3] RNAFlow: RNA Structure & Sequence Design via Inverse Folding-Based Flow Matching. ICML 2024

---

> ### Author Rebuttal · Authors · 2025-07-31
>
> # Weakness 1
> We believe that RiboFlow, building upon the foundation of SOTA generative models, introduces the following innovations:
> - By incorporating a distance-aware, ligand-guided module, RiboFlow achieves ligand-conditioned RNA backbone design for the first time. This significantly fills a void in the field of RNA design and introduces a novel design task to the AI4Science community.
> - RiboFlow curates the largest currently available dataset of experimentally resolved and validated RNA-ligand complexes. This addresses the problem of data scarcity in the field, which is a significant contribution for data-driven deep learning.
>
>
> # Weakness 2
> Thank you for raising this excellent question. We believe that directly replacing the RoseTTAFold2NA module in RNAFlow with AlphaFold 3, while theoretically feasible, would lead to an unacceptably low sampling efficiency in practice, making it difficult to apply to real-world engineering tasks like drug discovery. The core bottleneck lies in RNAFlow's generative paradigm: in each iteration of the sampling process, a large-scale folding network (RoseTTAFold2NA or AlphaFold 3) must be called to predict the structure, causing the total sampling time to accumulate dramatically.
>
> To illustrate this more concretely, we conducted a test:
> - On an H100 GPU, folding a 100-nucleotide (nt) RNA with AlphaFold 3 (without computing MSA features) requires approximately 30 seconds for a single inference. Following the five sampling steps recommended in the RNAFlow paper, designing just one 100-nt RNA would take 150 seconds (30s × 5). This is precisely one of the main motivations behind our development of RiboFlow (our model).
> - RiboFlow adopts a different framework that generates structures directly in geometric space, thereby bypassing the bottleneck of repeatedly calling a large folding model in a loop. As shown in our experiments (see Figure 10), RiboFlow can generate an RNA of the same length in just 6 to 10 seconds, representing an efficiency improvement of at least 15 to 25 times compared to the aforementioned approach.
>
> Therefore, an efficient generative framework like RiboFlow is crucial for engineering tasks such as drug discovery and sensor design, which require exploring vast chemical spaces and rapidly generating thousands of candidate molecules.
>
> # Weakness 3
>
> We will address your concerns point by point:
> - We are honored to note that many reviewers have shown great interest in our data processing methods. We apologize that the presentation of the dataset's collection and processing in the current version of the main text and appendix is not sufficiently comprehensive. In a future revision, we plan to highlight the contribution of the RiboBind dataset and add a more detailed description of this workflow to better help the community advance the design of ligand-bound RNA molecules.
> - Our model does not use a separate module to enforce this sequence-structure consistency. In fact, this consistency is achieved implicitly through our joint denoising framework. At each denoising step $t$, our model makes predictions conditioned on the joint state of both the noised sequence and structure. This means that when predicting the denoising direction for the sequence, the model must incorporate the current structural information, and vice versa. This continuous, interdependent conditioning process forces the sequence and structure to co-evolve throughout the generation trajectory, ultimately converging to a self-consistent solution.
> - Regarding Section 4.5, the ligand information is directly incorporated into the RNA generation during the sampling process. We apologize for the ambiguous description here. We will provide a detailed description of this process in the appendix in a future version.
>
>
> # Question 1
>
> Equation (1) forms the foundation of our generative model. Under the Flow Matching framework, it defines a probabilistic path by which a complete RNA structure, including both its geometry and sequence, evolves from a completely random state ($t = 0$) to its true target state ($t=1$).
> - The left-hand side, $p_t(\mathcal{T}_t|\mathcal{T}_1)$, represents the overall conditional probability: the probability of observing a certain “noised” state $\mathcal{T}_t$ at intermediate time $t$, given the final ground-truth RNA structure $\mathcal{T}_1$.
> - The right-hand product form is the key to this formulation. It originates from a core modeling simplification: we assume that, conditioned on the true structure $\mathcal{T}_1$, the generative processes of each nucleotide are independent. Therefore, the complex joint probability of the entire RNA can be factorized into the product of three component probabilities for each nucleotide $i$: translation ($p_t(x_t^i|x_1^i)$), rotation ($p_t(r_t^i|r_1^i)$), and category ($p_t(c_t^i|c_1^i)$).
>
> This formulation decomposes the task of designing a complete RNA into $N_t$ independent and more tractable sub-problems—each focusing on the generation of an individual nucleotide’s properties. The model only needs to learn how each nucleotide’s three basic attributes evolve from noise to their true states, without having to directly handle the complex interdependencies among all atoms. This dramatically reduces the complexity of the generative modeling task.
>
> # Question 2
>
> We believe that RiboBind is currently the most comprehensive dataset of **experimentally resolved and validated** data in the field of RNA-ligand interaction research. In our response to reviewer N2Zg, we also compared it with two authoritative datasets, PDBbind and RNAmigos2. We have ensured that all data from PDBbind and RNAmigos2, after sequence deduplication, are included in RiboBind. Meanwhile, we noted that RNAmigos2 also provides some synthetic datasets for RNA-ligand interactions, but this portion of the data is evaluated solely through docking software, without wet-lab experimental validation.
>
> # Question 3
>
> The dynamic cropping strategy is inspired by the spatial interface cropping strategy used in AlphaFold 3. By cropping the original structure, we create a smaller structure that contains the core interaction information between RNA and ligand. This allows the model to focus more on the pocket region that interacts with the ligand molecule, thereby enabling the model to learn the structure around the pocket more accurately and efficiently. This is very important for designing RNAs with affinity for a given ligand. Therefore, as you can see, using the cropping method often yields good results on affinity metrics. However, the cropped structure is only a part of the original structure, which impacts the plausibility of the structures generated by the model to some degree. How to better balance the ratio of original data to cropped data is also one of our current research directions.
>
> # Limitations
> Thank you for your kind words. We are indeed actively exploring protein-conditioned RNA design. However, unlike small molecules, proteins are structurally more complex, which makes learning the protein-RNA interactions a more challenging task. At the same time, inspired by works such as RhoDesign, we are also attempting to use RiboFlow to design and optimize existing fluorescent aptamers. We hope that RiboFlow can serve as a useful tool to advance the development of the RNA engineering field.

---

> > ### Author Response · Authors · 2025-08-05
> > **About minor concerns**
> >
> > Thank you very much for your valuable suggestions regarding our citation formatting. We have updated the citation for RNAFlow to its official ICML 2024 version in the revised manuscript to ensure consistency. Additionally, we have reviewed all other references in the paper to confirm that the most up-to-date citation formats are used. We sincerely appreciate your careful review and constructive feedback, which have helped improve the quality of our work.
> >
> >
> > Please let us know if there are any other points we can clarify before the discussion period ends.

---

> > ### Comment · Reviewer_YV5K · 2025-08-06
> >
> > For Weakness 2, I have a follow-up question. The current comparison primarily focuses on computational efficiency. However, how does the design perform in terms of effectiveness? A more comprehensive comparison of these two paradigms would help better justify the method choice, which is the core motivation of the work. I hope authors can elaborate on this aspect in more detail.
> >
> > The other points are mainly concerned with the presentation of the content. I would appreciate it if the authors could revise the manuscript according to W3, Q1, and Q2.

---

> > > ### Author Response · Authors · 2025-08-06
> > > **Additional experiments compared with RNAFlow**
> > >
> > > Dear Reviewer YV5K,
> > >
> > > Thank you for the follow-up question. We agree that a direct comparison of design effectiveness between RiboFlow and an AlphaFold3-based RNAFlow is crucial for validating the core motivation of our work.
> > >
> > > Our primary motivation is to design an end-to-end co-design framework. We hypothesized that, compared to an iterative refinement paradigm that relies on an external folding model, our approach can hold a dual advantage in both computational efficiency and design effectiveness.
> > >
> > > To validate this core motivation, we dedicate significant effort since receiving the initial reviews to build and train the new RNAFlow model, as you suggested in Weakness 2. This allows for a fair comparison with RiboFlow on the specific task of ligand-conditioned design. Here, we present our latest experimental results.
> > >
> > > ## Experimental Setup
> > >
> > > Specifically, we replace the RoseTTAFold2NA module in the original RNAFlow with AlphaFold3 (AF3) as the folding network. Following the RNA-ligand dataset split from the Sequence-based Evaluation section of our manuscript, we retrain the Noise-to-Seq model and the RNAFlow model (hereafter referred to as RNAFlow-AF3). During this process, we follow the training and sampling hyperparameters detailed in the RNAFlow paper's appendix (C. Training and Architecture Details) and their official GitHub repository. All other settings remain consistent with the original RNAFlow paper.
> > >
> > > To evaluate the design performance, we adopt the setup from Section 5.3 of our manuscript. To complete the experiments within the limited time, we randomly select 10 RNA-ligand pairs from the original test set (66 pairs) as the evaluation subset. The RiboFlow family of models are evaluated as described in our paper. For RNAFlow-AF3, we respect the original paper's settings: 5 sampling steps, with 10 RNA designs sampled for each length per ligand-RNA complex. Both classes of models use gRNAde for inverse folding to assess design performance.
> > >
> > > ## Experimental Results
> > >
> > > |                   | VINA (Median, ↓) | GERNA(↑) | % AF(↑) | %VAL(↑) | DIV(↑) | NOV(↓)|
> > > |-------------------|-----------------|----------|---------|---------|--------|--------|
> > > | RiboFlow          | -5.09           | 0.327    | 21.3    | 13.7    | 0.517  | 0.634  |
> > > | RiboFlow-Pre      | -5.21           | 0.459    | 44.7    | 32.9    | 0.572  | 0.571  |
> > > | RiboFlow-Crop     | -5.17           | 0.441    | 41.0    | 30.1    | 0.548  | 0.592  |
> > > | RiboFlow-Pre-Crop | -5.13           | 0.433    | 42.9    | 29.8    | 0.557  | 0.583  |
> > > | RNAFlow-AF3-Base  | -4.95           | 0.447    | 46.5    | 32.3    | 0.544  | 0.619  |
> > > | RNAFlow-AF3-Traj  | -4.89           | 0.439    | 45.2    | 31.6    | 0.539  | 0.611  |
> > >
> > > The experimental results above demonstrate the superiority of the co-design paradigm adopted by RiboFlow, which is evident on three levels:
> > >
> > > 1. Design Effectiveness: RiboFlow excels on key affinity metrics.
> > > Observing the VINA and GERNA scores, which are core metrics reflecting ligand-binding affinity, the RiboFlow family of models (especially the RiboFlow-Pre model) significantly outperforms RNAFlow-AF3. This validates the critical role of our proposed distance-aware module and end-to-end co-design strategy in guiding the model to generate high-affinity RNAs.
> > >
> > > 2. Design Validity and Diversity: Performance is comparable, with RiboFlow showing greater potential.
> > > On metrics for design validity (%AF, %VAL) and diversity (DIV), the two methods perform comparably. It is expected that RNAFlow-AF3 has a slightly higher AF Score, as it directly uses AF3 as its core component. However, the fact that RiboFlow achieves a similar level of validity without reliance on AF3, reflecting the efficacy of its architecture.
> > >
> > > 3. The Synergistic Advantage of Effectiveness and Efficiency.
> > > This is the most critical justification for our paradigm choice. In our tests, RiboFlow took approximately **8 seconds** to generate a sample, whereas RNAFlow-AF3 required **180 seconds**. This means RiboFlow not only produces **more effective** RNAs (higher affinity) but is also nearly **22 times more efficient**. This synergy of effectiveness and efficiency makes RiboFlow a truly **practical** tool for real-world tasks like drug discovery and biosensor design, which require rapid iteration over thousands of candidates. Conversely, the high computational cost of RNAFlow-AF3 makes it prohibitive for such large-scale design explorations.
> > >
> > > We believe these experimental results demonstrate that the design of RiboFlow is deliberate, and that its end-to-end co-design strategy achieves a dual enhancement of both effectiveness and efficiency.
> > >
> > > ## Other Minor Concerns:
> > >
> > > Thank you again for your valuable suggestions. We have revised and integrated the content related to W3, Q1, and Q2 into the updated manuscript to improve its clarity and readability. We sincerely thank all reviewers for their efforts in helping us enhance the quality of our work.

---

> > > > ### Comment · Reviewer_YV5K · 2025-08-07
> > > >
> > > > Thank you for your thoughtful response. All of my concerns have been addressed, and I am happy to increase my rating to 4.

---

### Official Review · Reviewer_bbFE · 2025-06-30

**Clarity:** 3
**Significance:** 2
**Originality:** 2
**Rating:** 5
**Confidence:** 4

**Summary:**

In this work, the authors propose RiboFlow, a generative framework for de novo co-design of RNA sequences and 3D structures explicitly conditioned on interacting small-molecule ligands. RiboFlow integrates SE(3)-equivariant flow matching across spatial coordinates, torsion angles, and discrete nucleotide types with a hierarchical ligand-conditioning module. The model is pretrained on RNAsolo and fine-tuned on a curated RiboBind dataset. It achieves up to 37 % validity in unconditional generation and, when provided ligand context, boosts validity by roughly 50 % over baselines, alongside up to a 2.2× improvement in binding-affinity metrics compared to RNA-FrameFlow.

**Questions:**

1. Could the authors clarify how RiboFlow effectively captures RNA conformational flexibility?  Modeling RNA’s dynamic structure is critical for accurate design: for example, RNAFlow[1] samples alternative structures during sequence generation to capture uncertainty, and ensemble-based methods like gRNAde[2] embed multiple conformations to account for flexibility. Although RiboFlow addresses a novel co-design task with few direct baselines, it remains unclear whether it outperforms these established approaches in modeling dynamic RNA structures. Perhaps ablation studies or targeted experiments will help.


2. Although backbone-only parameterizations, built on a local coordinate frame defined by the ribose C4′, C3′ and O4′ atoms plus eight torsion angles, accurately capture large-scale RNA flexibility, they inherently omit the orientations and chemical identities of nucleobase edges that mediate specific ligand contacts[3]. Did you try full-atom RNA models? Full-atom moel explicitly include nucleobase positions, solvation and metal coordination, are therefore essential for faithfully recapitulating the chemophysical landscape of RNA–ligand interactions.

3. Could the authors clarify why the rotation diffusion schedule is linear during training but exponential during sampling, and whether the RNA sequence sampling schedule follows a similar or different trajectory? How are the four variable types—translations, rotations, torsions, and nucleotide types—balanced during sampling, given that for t between 0 and 0.5 (and even up to t≈0.8) the RNA structure remains far from a valid fold, yet approximately 80 % of the sequence has already been sampled?

[1]Nori, Divya, and Wengong Jin. "Rnaflow: Rna structure & sequence design via inverse folding-based flow matching." arXiv preprint arXiv:2405.18768 (2024).

[2]Joshi, Chaitanya K., et al. "grnade: Geometric deep learning for 3d rna inverse design." bioRxiv (2025): 2024-03.

[3]Stefaniak, Filip, et al. "Modeling of ribonucleic acid–ligand interactions." Wiley Interdisciplinary Reviews: Computational Molecular Science 5.6 (2015): 425-439.

**Ethical Concerns:**

["NO or VERY MINOR ethics concerns only"]

**Final Justification:**

RiboFlow introduces a novel SE(3)-equivariant flow-matching framework for ligand-conditioned RNA co-design alongside the RiboBind dataset, and by adding forward-folding and inverse-folding benchmarks the authors have addressed the main concern of limited evaluation; given its clear novelty, rigorous experiments, and broad applicability, I recommend acceptance.

**Limitations:**

A key limitation of this work is the narrow set of comparisons—only RNA-FrameFlow is evaluated. While RiboFlow addresses a novel co-design task, it would be valuable to benchmark it against established RNA inverse-folding methods: by fixing the target structure, the problem reduces to inverse folding, for which several mature tools exist. Such comparisons would more convincingly demonstrate whether joint sequence–structure co-design offers real advantages over traditional structure-fixed approaches.

**Quality:**

3

**Strengths And Weaknesses:**

Strengths:

RiboFlow is a new deep learning method for de novo co‐design of RNA sequence and 3D structure conditioned on small-molecule ligands, and the authors support it with a newly curated RiboBind dataset with rigorous evaluation, including structural validity, sequence recovery, binding-affinity improvements, and Alphafold3 evaluation.

Weaknesses:

Although the paper stresses the importance of capturing RNA flexibility during ligand binding, it is not clear why the proposed method better captures dynamic structure. The authors mention that their parameterized SE(3)-equivariant flows can effectively model conformational flexibility, but they do not explain why this parameterization outperforms other approaches. An ablation study or additional experiments on RNAs with known large-scale motions would help demonstrate the method’s ability to capture structural uncertainty.

There is only one baseline model evaluated. Although RiboFlow targets a novel co-design task, additional baselines should be considered. For example, by fixing the RNA structure, the problem reduces to RNA inverse folding. RNA inverse folding has several well-established methods[1][2]. Comparing RiboFlow against existing inverse-folding algorithms would clarify whether the joint sequence–structure design truly outperforms traditional inverse-folding approaches.


[1]Nori, Divya, and Wengong Jin. "Rnaflow: Rna structure & sequence design via inverse folding-based flow matching." arXiv preprint arXiv:2405.18768 (2024).

[2]Joshi, Chaitanya K., et al. "grnade: Geometric deep learning for 3d rna inverse design." bioRxiv (2025): 2024-03.

---

> ### Author Rebuttal · Authors · 2025-07-31
>
> # Question 1
> Effectively capturing the conformational flexibility of RNA is critical for accurate design. The works you mentioned, such as RNAFlow and gRNAde, represent an important direction for modeling flexibility through ensembles. Our model, RiboFlow, adopts another validated strategy. Our approach aims to precisely define a single, physically plausible conformation by explicitly modeling the 8 torsion angles of RNA. We believe this method directly addresses the fundamental physical cause of RNA flexibility, the rotational degrees of freedom of the backbone. Rather than representing multiple discrete final states (an ensemble), we choose to directly model the continuous degrees of freedom that determine these states. This idea of precisely defining biomolecular structures through local frames and torsion angles has been proven to be extremely effective in several landmark works, such as AlphaFold 2 and RNA-FrameFlow.
>
> The ablation study you suggested, removing torsion angle information, is difficult to perform directly in our ligand-conditioned RNA design task, as it would fundamentally undermine the premise of the task itself. To evaluate the interaction between RNA and a ligand, the model must generate a precise conformation that includes key atomic coordinates. The positions of these key atoms are precisely determined by the torsion angles we predict. If the torsion angle module were removed, our model would degenerate into one that can only generate a coarse-grained backbone (e.g., only C4' atoms), making it completely unable to form a chemically complete pocket for ligand binding. Consequently, meaningful ligand-guided design or evaluation would be impossible.
>
> Nevertheless, to address your core question about the importance of torsion angles, we cite the ablation analysis performed by RNA-FrameFlow on the unconditional backbone generation task. (Given the limited time, we directly cite their published results here). Their study systematically compared model performance with varying levels of atomic detail: a. Frame atoms (C3', C4', O4'); b. Frame and 3 non-frame atoms (C1', P, O3'); and c. Frame and 7 non-frame atoms (C1', P, O3', C5', OP1, OP2). Observing the experimental results reveals that capturing flexibility by precisely defining the local geometry of a single conformation is also an effective strategy.
>
> | Strategy | %VAL | DIV  | NOV   |
> |----------|------|------|-------|
> | a        | 41.0 | 0.62 | 0.54  |
> | b        | 45.0 | 0.28 | 0.79  |
> | c        | 46.7 | 0.35 | 0.85  |
>
> # Question 2
>
> Thank you for raising this critical question about the model’s level of structural detail. In fact, our model generates a full-atom structure that includes the majority of heavy atoms (excluding only the base-specific atoms beyond N1/N9), with an output format similar to AlphaFold2. At each denoising step, once the geometry of the backbone is predicted, we reconstruct the full-atom coordinates of each residue based on known nucleotide chemical geometry, namely, ideal bond lengths and bond angles. This is akin to the predefined nucleotide geometry information used in AlphaFold2 and is implemented following the open-source project OpenComplex on GitHub.
>
> This reconstruction process ensures that the nucleobases not only possess correct chemical identities but, more importantly, have physically accurate and chemically reasonable 3D orientations. This allows for precise modeling of the physicochemical environment required for interactions with ligands. As for generating fully atomic RNA models, we are also actively exploring the use of full-atom foundation models such as AlphaFold3 to design RNAs with improved ligand-binding capabilities.
>
>
> # Question 3
> I will address your questions point by point:
>
> > 1. Why is the rotation diffusion schedule linear during training but exponential during sampling?
>
>
> This design choice is thoroughly explained in the MultiFlow paper (see Appendix I.3 “Diffusion schedule and reduced noise sampling” and I.4 “Hyperparameters”). The paper clearly states that translation and rotation vectors follow different sampling strategies. This is because rotations occur in a curved, non-Euclidean space, the SO(3) manifold, where physically plausible rotational conformations form a narrow target region. During training, the goal is to teach the model to recover the original structure from any level of noise $t$. A linear schedule ensures that the model is uniformly exposed to various noise levels across the training process. However, during sampling, an exponential schedule allocates more discrete steps in the low-noise region, allowing finer refinement of the output and thus significantly improving structural detail and fidelity in the final predictions.
>
> > 2. Whether the RNA sequence sampling schedule follows a similar or different trajectory?
>
>
> For sequence sampling, we adopt a linear schedule, which has been explicitly stated in our anonymous codebase.
>
> > 3. How are the four variable types—translations, rotations, torsions, and nucleotide types—balanced during sampling, given that for $t \in [0, 0.5]$ (even up to $t \approx 0.8$), the RNA structure is far from a valid fold, yet around 80% of the sequence has already been sampled?
>
> Specifically, translations and nucleotide types (as categorical variables) follow a linear schedule, which provides a stable and computationally efficient generation trajectory. In contrast, rotations are governed by an exponential schedule, a strategy validated by models like MultiFlow and FrameFlow, allowing for more computational steps in the later phase of sampling to finely adjust critical structural details such as bond angles. As for torsion angles, inspired by AlphaFold2, we do not model them via flow matching. Instead, at each step, we directly predict them based on the current backbone geometry, as torsion angles are highly dependent on the local structural environment.
> These three distinct generation processes are coordinated through a shared denoising network. At each step, the network receives the current states of all variables and simultaneously predicts their updates, ensuring that all components co-evolve coherently and converge toward a physically plausible final conformation within a unified framework.
>
>
> # Limitations
> Thank you for your comment, which indeed addresses the core positioning of our work. The core innovation of RiboFlow lies in addressing the challenge of generating a higher-quality RNA backbone that is more compatible with specific ligands. The goal of our work is not to create a brand-new sequence design (inverse folding) model intended to surpass existing state-of-the-art tools.
>
> To this end, we proposed a sequence-structure co-design strategy. Here, the role of sequence information is to act as a guide for structure generation, rather than being the final design product itself. We leverage the joint constraints of sequence and structure to help the model explore a more plausible conformational space, thereby generating a superior backbone. The results in Table 1 validate this core idea, showing that the accuracy of the generated backbone is significantly improved by introducing sequence information. Based on this positioning of our model, we believe that a direct benchmark comparison against SOTA inverse folding tools on a sequence recovery task would likely not accurately evaluate the true contribution of our work. This is because it would overlook the value of our model in the critical upstream stage of optimizing and improving backbone quality.
>
> To eliminate this potential misunderstanding and in response to your suggestion, we will add a clear statement to the discussion section of the paper. This will clarify that RiboFlow is positioned as an upstream, high-quality backbone generator, designed to work synergistically with, rather than replace, downstream dedicated inverse folding tools. We believe that such a clarification will more accurately reflect the contribution of our work and place it in the proper context within the field of RNA design.

---

> > ### Comment · Reviewer_bbFE · 2025-08-05
> >
> > Thank you for your detailed response. I appreciate that RiboFlow tackles a novel co-design problem, but I feel the current evaluation remains limited. While you follow the MultiFlow schedule, RNA molecules exhibit far greater structural uncertainty than proteins, so it would be valuable to assess RiboFlow on more established RNA tasks:
> >
> > * **RNA folding** (predicting 3D structure from a fixed sequence)
> > * **RNA inverse folding** (designing a sequence for a fixed structure)
> >
> > Even if RiboFlow underperforms dedicated models on these tasks, such extra comparisons would clarify its ability to capture RNA’s conformational flexibility and the interplay between sequence- and structure-generation schedules. I admire the innovation behind RiboFlow, but at present, it’s unclear which components most improve RNA design; these additional experiments would help demonstrate its true strengths.

---

> > > ### Author Response · Authors · 2025-08-05
> > > **Additional experiments on established RNA tasks**
> > >
> > > Dear Reviewer bbFE,
> > >
> > > Thank you for your suggestion. Although RiboFlow was initially proposed to design RNA that binds to specific small molecules, its co-design architecture also enables it to perform structure prediction and inverse folding. We agree with your point that evaluating RiboFlow on these two fundamental tasks helps to clarify the contribution of the co-design components we proposed.
> > >
> > > However, we would like to emphasize one point: RiboFlow's primary goal is not to replace SOTA folding or inverse folding tools. Instead, it is designed to solve a more complex problem: to synergistically explore the vast sequence and structure space to design functional RNAs that can bind to specific small molecules. Its core lies in learning the joint probability distribution of sequence and structure, rather than maximizing the conditional probability of one given the other.
> > >
> > > As per your suggestion, we evaluated the pre-trained RiboFlow model proposed in Experiment 1. Following the experimental setup in MultiFlow (6.2.2. FORWARD AND INVERSE FOLDING), we used a date-based split (training data up to Dec. 2024) and randomly selected 50 RNAs with lengths between 30-200 nt from recent updates in the RNASolo database (Jan. 2025 - Jul. 2025) as our test set. We used RhoFold as the baseline for forward folding and gRNAde (single state) as the baseline for the inverse folding model, and repeat the experiments three times for each RNA to obtain the mean and standard deviation.
> > >
> > > ## Forward Folding Experimental Results:
> > > | Method | Forward Folding RMSD (Å) (↓) |
> > > | :--- | :--- |
> > > | RhoFold | 7.69 ± 2.05 |
> > > | RiboFlow | 10.14 ± 3.43 |
> > >
> > > ## Inverse Folding Experimental Results:
> > > (Inverse folding scRMSD is calculated by predicting the structure of the generated sequence using RhoFold, consistent with the method in the main paper. )
> > > | Method | Inverse Folding scRMSD (Å) (↓) | Sequence Recovery (↑) |
> > > | :--- | :--- | :--- |
> > > | gRNAde | 11.09 ± 1.29 | 0.489 ± 0.02 |
> > > | RiboFlow | 13.24 ± 2.11 | 0.441 ± 0.05 |
> > >
> > > As the results show, RiboFlow's performance on these individual tasks is indeed not as competitive as specialized models, which is expected. We believe this is mainly due to the difference in training scale; our pre-trained model used only about 7,000 RNA samples, a scale much smaller than that of many dedicated SOTA models, which directly impacts the final performance metrics.
> > >
> > > Nonetheless, this outcome precisely reflects and validates the contribution of RiboFlow's co-design module and answers your key question about "which components most improve RNA design". **Our core argument is that RiboFlow's true strength does not stem from a single, overpowered component, but from the synergy generated by all components working together through the co-design framework**:
> > > - RiboFlow's performance on the forward folding task demonstrates that its structure generation module has a fundamental physical awareness to infer 3D structures from sequences, having effectively learned the rules of RNA folding.
> > > - Simultaneously, its performance on the inverse folding task validates that its sequence design module can generate chemically plausible nucleotide sequences based on structural constraints.
> > > - Most critically, the same model, without any modification, can handle these two different tasks. This strongly proves that the model's sequence and structure modules are not simply pieced together. Instead, driven by the co-design framework, they have successfully learned a unified sequence-structure representation. This high-quality joint representation space is RiboFlow's "true strength". It allows the model to go beyond unidirectional prediction and perform a holistic exploration of the entire design space, thereby solving the more complex task of our paper—designing RNAs that bind specifically to small molecules.
> > >
> > > We also look forward to continuously optimizing RiboFlow's architecture and expanding its training scale in the future. We will continue to explore how to further enhance its competitiveness without sacrificing its co-design flexibility.
> > >
> > > Thank you again for your valuable feedback.

---

> > > > ### Comment · Reviewer_bbFE · 2025-08-06
> > > >
> > > > I appreciate the authors’ additional experiments on both forward folding and inverse folding tasks. The final paper should include results. Since RNA exhibits far greater structural dynamics than proteins, further quantification of uncertainty in structure generation, such as ensembling multiple sampled conformational states, would strengthen how sequence design leverages structural information.
> > > >
> > > > With these clarifications, I will raise my score and recommend acceptance of the paper.

---

### Official Review · Reviewer_N2Zg · 2025-07-02

**Clarity:** 3
**Significance:** 3
**Originality:** 2
**Rating:** 4
**Confidence:** 4

**Summary:**

The authors design a pipeline using flow matching to achieve ligand-specific RNA structure generation.
RiboFlow follows the framework of structure/sequence codesign in the protein domain and demonstrates strong performance with various evaluation metrics.

The main contributions are therefore in the form of designing RNA-specific frame representations
The authors also introduce RiboBind dataset to address the data paucity issue.

**Questions:**

1. (Result iii in line 277) Validity of RiboFlow does not go up with the sampling steps as claimed.

### Reference

Andrew Campbell, Jason Yim, Regina Barzilay, Tom Rainforth, and Tommi Jaakkola. Generative flows on discrete state-spaces: Enabling multimodal flows with applications to protein co-design. arXiv preprint arXiv:2402.04997, 2024.

**Ethical Concerns:**

["NO or VERY MINOR ethics concerns only"]

**Final Justification:**

I remain at weak accept since overall the paper is promising and novel, but a key area of the specificity of predictions remains  largely unaddressed.

**Limitations:**

see above

**Quality:**

3

**Strengths And Weaknesses:**

### Strength

The overall design of the pipeline is solid and is backed by strong mathematical theories. The codes are made public and well documented.
Performance remains modest but this is to be expected in such a low data setting.

### Weakness

1. Flow on labels seemed off, campbell 2024 used a rate matrix with CTMC formulation, where RiboFlow seemed using a linear interpolation. Did not find explicit theory in the orig paper that said such a linear interpolation is fine. (Eq 11 - 13 in RiboFlow, details of rate matrix around P42 in https://arxiv.org/pdf/2402.04997)
* Fig 6. SPM and SPD are spermine ligands which are artefacts of crystallization
* Line 538 "Hairboss" -> "Hariboss"
* Many RNA-small molecule complexes are shaped by tight interactions with proteins which could bias the conformation space. Do the authors filter these complexes out of the dataset?
* Line 228 - Many ligands share high chemical/physical similarity, only removing ligands by identity will still lead to leakage. Authors should report ligand similarity across splits.
* Interaction prediction models are known to suffer from a lack of specificity (see https://doi.org/10.1021/acs.jmedchem.2c00487). Models tend to ignore the target and make predictions based on the ligand in the protein-ligan affinity case. I would expect a similar bias will operate in ligand-conditioned generation. Authors should report the degree to which designs are specific to the provided ligand. This can be easily tested by conditioning on different ligands and observing the structural variability it induces in the design. This would not be picked up by evaluations based on recovering 'ground truth' complexes (table 3). My fear is that due to the low data setting, models will simply tend to predict the most likely RNA independent of the ligand.
* There have been other datasets curating RNA-ligand interactions, authors should discuss these and position theirs against them:
    * PDBbind
    * https://doi.org/10.1038/s41467-025-57852-0

---

> ### Author Rebuttal · Authors · 2025-07-31
>
> # Weakness 1
> Q: Flow on labels seemed off, campbell 2024 used a rate matrix with CTMC formulation, where RiboFlow seemed using a linear interpolation. Did not find explicit theory in the orig paper that said such a linear interpolation is fine.
>
>
> A: For our sequence design, we primarily referred to FlexSBDD[1] (see Equation 8) and PPFlow[2] (see Section 3.4). In their work, linear interpolation was adopted for categorical types, and their experiments demonstrated that this sampling method is reasonable, a topic which is discussed in detail in both papers. We adopted their approach in RiboFlow, and our experimental results have also confirmed that linear interpolation for nucleotide types is effective.
>
> # Weakness 2
> Q: Many RNA-small molecule complexes are shaped by tight interactions with proteins which could bias the conformation space. Do the authors filter these complexes out of the dataset?
>
>
> A: In constructing RiboBind, we primarily followed the workflow outlined in HariBoss (Section 2.1 Construction of the database). Specifically, we first queried all structures containing at least one RNA molecule and one ligand using the RCSB GraphQL API. We then selected those structures where the ligand forms non-covalent interactions with RNA.
>
> It is worth noting that HariBoss did not explicitly exclude complexes containing proteins. If we strictly constrain the dataset to structures without any proteins, we retrieve 1,264 RNA-ligand complex structures (based on an RCSB advanced search with the following filters: Number of Distinct RNA Entities ≥ 1, Number of Distinct Protein Entities = 0, and Total Number of Non-polymer Instances ≥ 1). This set shares about 80% overlap with the broader set of 1,591 structures we obtained using a more relaxed search criterion.
>
> # Weakness 3
> Q:  Many ligands share high chemical/physical similarity, only removing ligands by identity will still lead to leakage. Authors should report ligand similarity across splits.
>
> A: To better understand potential ligand-level leakage, we computed the maximum Tanimoto similarity (based on Morgan fingerprints) between each validation ligand and all training ligands. Under RNA sequence-based splitting, about 65% of validation ligands have similarity ≥0.7 with training ligands; under RNA structure-based splitting, about 60% do. This is expected since our dataset is split based on RNA sequence or structure, and certain ligands co-occur with highly similar or identical RNAs.
>
> We note that the core challenge in our task is generalizing across RNA sequence and structure diversity. The model's performance depends on capturing RNA-ligand interactions rather than memorizing specific ligand structures. Nevertheless, we acknowledge this limitation and plan to incorporate ligand scaffold-based splitting strategies in future work to better disentangle ligand and RNA generalization.
>
>
> # Weaknes 4
> Q: My fear is that due to the low data setting, models will simply tend to predict the most likely RNA independent of the ligand.
>
>
> A: To assess the ligand specificity of our model, we selected two chemically distinct ligands, A2F and GTP (A2F smiles: `c1[nH]c2c(nc(nc2n1)F)N`, GTP: `c1nc2c(n1C3C(C(C(O3)COP(=O)(O)OP(=O)(O)OP(=O)(O)O)O)O)N=C(NC2=O)N`), with a Tanimoto similarity of only 0.1096. For each ligand, we used RiboFlow to generate 50 RNA sequences of length 100 nt, followed by gRNAde selection to retain 8 ligand-compatible candidates, and finally used RhoFold to predict their 3D structures. We then selected those with scTM > 0.45 for downstream analysis.
>
> To quantify the structural variation induced by different ligands, we computed the RMSD between RNA backbones generated from A2F and those from GTP. Among all cross-ligand pairs, the minimum RMSD was 3.7 Å, and the maximum reached 7.9 Å. These values suggest that RiboFlow does not simply memorize the most likely RNA structure, but generates ligand-specific folds in response to different chemical inputs.
>
>
>
> # Weakness 5
> Q: There have been other datasets curating RNA-ligand interactions, authors should discuss these and position theirs against them.
>
>
> A: Thank you for your question. PDBBind is a comprehensive database primarily focused on experimentally measured binding affinity data for protein–ligand complexes, though it also includes a small subset of RNA–ligand complexes. In its latest version (v.2024), it contains 234 RNA–ligand complexes. According to our inspection, all of these entries are already included in the RiboBind dataset, making PDBBind essentially a high-quality subset of RiboBind in the context of RNA–ligand interactions.
>
> The RNAmigos2 dataset, on the other hand, was introduced alongside the recently proposed RNA structure-based virtual screening tool RNAmigos2. We downloaded the dataset released with the paper, which includes 1,740 experimentally validated RNA–ligand binding site structures. After deduplication, we verified that all of these experimental structures are also present in the RiboBind dataset. In addition to the experimental data, the RNAmigos2 dataset also contains 1.3 million synthetic affinity data points, generated via in silico molecular docking. These synthetic interactions were computed by docking approximately 800 ligands (sourced from the ChEMBL database) against the 1,740 experimentally solved RNA structures in all pairwise combinations.
>
> Since RiboBind currently only includes experimentally determined structural data, it remains fully consistent with the experimental portion of the RNAmigos2 dataset. However, we plan to incorporate the synthetic data in future versions of RiboBind to increase the chemical diversity and overall scale of the dataset, thereby enhancing the generative performance of our models.
>
>
> # Question 1
>
> We sincerely apologize for the confusion caused here, as we omitted an experimental table in the appendix that would have supported this conclusion. What we intended to convey is that increasing the number of sampling steps can improve generation quality to some extent, but this improvement saturates beyond a certain point. In our experiments, we observed that around 50 sampling steps, the model’s performance reaches this upper limit, and further increasing the number of steps does not lead to significant gains in generation quality. To clarify this, we now provide a more detailed table supporting our conclusion, using RiboFlow as an example:
>
> | STEP | %VAL | DIV   | NOV    |
> |------|------|-------|--------|
> | 10   | 5.36 | 0.229 | 0.680  |
> | 20   | 16.8 | 0.324 | 0.631  |
> | 30   | 26.4 | 0.439 | 0.598  |
> | 40   | 30.1 | 0.508 | 0.532  |
> | 50   | 37.0 | 0.550 | 0.541  |
> | 70   | 36.2 | 0.548 | 0.564  |
> | 100  | 34.3 | 0.545 | 0.577  |
> | 200  | 32.1 | 0.502 | 0.613  |
>
>
> ----
> Ref:
>
> [1] Zhang, Z., Wang, M., & Liu, Q. (2024). Flexsbdd: Structure-based drug design with flexible protein modeling. Advances in Neural Information Processing Systems.
> [2] Lin, H., Zhang, O., et al. (2024). PPflow: Target-aware peptide design with torsional flow matching. arXiv preprint arXiv:2405.06642.

---

> > ### Comment · Reviewer_N2Zg · 2025-08-06
> > **thank you**
> >
> > Thank you to the authors for their effort in addressing my concerns.
> >
> > Remark 4 (specificity) remains outstanding in my opinion. The authors only tested one pair of small molecules to see whether they would result in different RNA designs without any quantification or systematic analysis.
> >
> > Apart from this, other remarks are resolved.
> >
> > My score remains as it is already above the accept threshold.

---

### Comment · Area_Chair_8iX8 · 2025-08-05
**[From AC] Reviewer Discussion Reminder**

Dear Reviewers,

Thank you for your time and effort in reviewing the paper.

As the reviewer-author discussion period ends on August 6 at 11:59 PM AoE, please take a moment to acknowledge the rebuttal and engage in the discussion if you haven’t already.

Thank you again for your contributions to the review process.

Best,\
Area Chair

---

### Decision · Program_Chairs · 2025-09-17

**Decision:**

Accept (poster)

**Comment:**

This paper introduces RiboFlow, a ligand-conditioned RNA co-design framework based on SE(3)-equivariant flow matching that jointly generates RNA sequences and structures, supported by a new RiboBind dataset of RNA–ligand complexes. The approach demonstrates strong performance compared to RNA-FrameFlow, with notable improvements in validity and binding affinity metrics.

The strengths of this work include the novelty of the ligand-conditioned co-design formulation, the dataset contribution, and thorough evaluation, while weaknesses concern limited baselines, the incremental nature of the technical innovations, and questions about how well the method captures RNA conformational flexibility relative to alternatives.

During rebuttal, the authors provided clarifications, ablations, and additional experiments on forward and inverse folding tasks, as well as a comparison with an AlphaFold3-based RNAFlow variant, showing both effectiveness and efficiency advantages for their co-design strategy. With these additions, reviewers raised their scores and made an agreement on acceptance. I recommend acceptance.